# Framework for assessing lateral flows and solute transport during floods in a conduit-flow dominated karst system using the inverse problem for advection-diffusion equations

Cybèle Cholet[1,2], Jean-Baptiste Charlier[2], Roger Moussa[3], Marc Steinmann[1], and Sophie Denimal[1]

[1]Chrono-Environnement, UMR 6249 UBFC/CNRS, University of Burgundy Franche-Comté, Besançon, 25000, France
[2]BRGM, 1039 rue de Pinville, F-34000 Montpellier, France
[3]INRA, UMR LISAH, 2 Place Pierre Viala, F-34060 Montpellier, France

*Correspondence to:* Cholet Cybèle (cybele.cholet@univ-fcomte.fr)

**Abstract.** The aim of this study is to present a framework giving new keys to characterize the spatio-temporal variability of lateral exchanges for flows and solutes in a karst conduit network during flood events, treating both phenomena, the diffusive wave equation and the advection-diffusion equation, with the same mathematical approach assuming uniform lateral flow and solutes. A solution of the inverse problem for the advection-diffusion equations is then applied on data from two successive gauging stations to simulate flows and solute exchange dynamics after recharge. The study site is the karst conduit network of the Fourbanne aquifer in the French Jura Mountains, which includes two reaches of 5-10 km characterizing the network from sinkhole to cave stream, and to the spring. The model is applied after separation of the base and the flood components on discharge and total dissolved solids (TDS) in order to assess lateral flows and concentrations and compare them to help identify water origin. The results showed various lateral contributions in space - between the two reaches located in the unsaturated (R1), and in both unsaturated and saturated zone (R2) - as well as in time, according to hydrological conditions. Globally, the two reaches show a distinct response to flood routing, with important lateral inflows on R1 and large outflows on R2. By combining these results with solute exchanges and the analysis of flood routing parameters distribution, we showed that lateral inflows on R1 are the addition of diffuse infiltration (observed whatever the hydrological conditions) and localized infiltration in the secondary conduit network (tributaries) in the unsaturated zone, except in extreme dry periods. On R2, despite inflows on the base component, lateral outflows are observed during floods. This pattern was attributed to the concept of reversal flows of conduit/matrix exchanges, inducing a complex water mixing effect in the saturated zone. From our results we build the functional scheme of the karst system. It demonstrates the impact of the saturated zone on matrix/conduit exchanges in this shallow phreatic aquifer, and highlights the important role of the unsaturated zone on storage and transfer functions of the system.

## 1 Introduction

Hydraulic transfers and solute transport processes in karst aquifers are known very complex due to the organization of underground void structures leading to preferential drainage axis through a conduit network embedded in a less permeable fissured

matrix (Kiraly, 2003; Ford and Williams, 2013). Flow processes are driven by the spatial variability of the geometrical elements that constitute the conduit network as full pipes, open channels, or pipe constrictions (Covington et al., 2009), leading to rapid transitions from free-surface to pressurized flows, after recharge events. In such heterogeneous media, transport parameters are also dependent on scale effect, as mentioned by Hauns et al. (2001) who showed that retardation factor is dominant at a local scale but vanishes at the benefit of an increase in dispersivity with increased distances. In addition to the interaction between conduit and matrix compartments (Martin and Dean, 2001; Binet et al., 2017), solute concentration evolutions in the conduit network are strongly influenced by the mixing with inflows from tributaries (Perrin et al., 2007) or with flooded areas (Dewaide et al., 2016). These last papers pointed out the various and complex lateral exchanges along the conduit network, which remains an open question calling for new tools to investigate it.

Natural and artificial tracers are commonly used in catchment hydrology to better understand the spatial variability of lateral exchanges in order to study flows and the corresponding solute concentrations between main channel and adjacent hydrological units. Ruehl et al. (2006) highlighted strong channel losses despite storage exchange fluxes and lateral inflows. By analysing exchanges between consecutive reaches in a mountainous headwater stream, Payn et al. (2009) emphasized the importance of the geomorphological context as a driver for the spatial variability of gaining and losing reaches. Moreover, they demonstrated that many of the studied reaches were concurrently losing and gaining. Szeftel et al. (2011) simulated exchanges using the OTIS model (Runkel and Chapra, 1993; Runkel, 1998), which is a 1-D finite-difference model solving solute transport in the streams accounting for transient storage. By testing the influence of different conceptualizations of hydrologic exchanges on the estimation of transient storage parameters, they showed the complexity to model tracer evolution, due to the difficulty to model and quantify spatial patterns of tracer concentrations and magnitudes of lateral inflows and outflows. The same model was recently used by Dewaide et al. (2016) to simulate lateral exchanges in karst conduits in both unsaturated ("river stretches") and saturated zones ("flooded area"). This study furthermore showed that the parametrization of solute transport in the system had to account for interactions within the saturated zone. However, all these works were designed for low flow periods and are therefore not suitable to investigate the temporal evolution of lateral flows during flood events in conduit-flow dominated karst systems with large-magnitude flash flows.

The Saint-Venant equations (SVE) may be used to assess hydrodynamic processes as they describe unsteady flow in partially filled conduits (Saint-Venant, 1871) and are generally used to simulate discharge in conduit-flow dominated karst aquifers. As the conduit flow in the unsaturated zone is mainly controlled by free-surface conditions, Manning's equations are favoured, although they under-estimate head losses due to turbulent flows (Jeannin and Marechal, 1995). Equations to be used for the saturated zone should be adapted for pressurized conditions, such as the Preissmann slot model in ModBraC (Reimann et al., 2011), or the Darcy-Weisbach equation in pipe-flow models like the Storm Water Management Model (SWMM) (Jeannin, 2001; Campbell and Sullivan, 2002; Peterson and Wicks, 2006; Chen and Goldscheider, 2014). However, application of these models is questionnable, because detailed information – often not available - on hydraulic parameters is required, particularly concerning the location and geometry of conduits. As a consequence, such physically-based models are often applied in a

degraded mode (Le Moine et al., 2008), and are sometimes over-parameterized even in case of relatively well-known conduit networks. An alternative approach to model hydraulic processes would be to test the ability of simplified SVE with parsimonious parameters. For that, the diffusive wave equation can be considered as a relevant simplification of the full SVE (Moussa and Bocquillon, 1996). This approach was used successfully by Charlier et al. (2015) to assess lateral flows in karst rivers with numerous lateral in- and outflows. It is therefore a promising alternative to the more complex approaches cited above.

In most of the practical applications, the acceleration terms in the Saint-Venant equations can be neglected, and consequently by combining the differential continuity equation and the simplified momentum equation, the Saint-Venant system is reduced to only one parabolic equation: the diffusive wave equation (DWE) (Moussa, 1996; Fan and Li, 2006; Wang et al., 2014). The two parameters of the equation, celerity and diffusivity, are usually taken as functions of the discharge. Methods based on the finite-difference discretization techniques are generally used to resolve this equation, but when using numerical solutions, questions of the construction of finite-difference systems, methods for solving them, their stability and their accuracy are encountered (Moussa and Bocquillon, 1996). However, if the two parameters celerity and diffusivity can be assumed constant, and in the particular case without any lateral flow, the diffusive wave equation has an analytical solution: the Hayami (1951) model. Moussa (1996) extended this analytical solution to the case of uniformly distributed lateral flow which can be either positive (lateral inflow) or negative (lateral outflow). In the following, the Moussa (1996) model allows to calculate the outflow using as input the inflow and the lateral flow uniformly distributed, and using the two parameters celerity and diffusivity. Moreover, Moussa (1996) proposed an analytical solution which enables to calculate the temporal distribution of the lateral flow (under the hypothesis of uniformly distributed flow) by an inverse problem approach using as input both the inflow and the outflow, and using the two parameters celerity and diffusivity. Contrary to classical modelling approach where the measured output hydrograph is used only to validate a model, the inverse problem developed herein uses all information available in both input/output data. In comparison to numerical methods, the advantage of the Hayami solution extended by Moussa (1996) is an easy-to-use analytical solution. The solution of the inverse problem proposed is part of the hydrological model MHYDAS (Distributed Hydrological Modelling of AgroSystems ; Moussa et al., 2002).

To model conservative solute transport along a 1D flow path, the advection-diffusion equation (ADE) is largely used in hydrology (Runkel, 1996; Baeumer et al., 2001) and karst hydrology (Hauns et al., 2001; Luhmann et al., 2012). Under steady-state flow conditions, and in agreement with the mass conservation law, solute transport can be expressed by ADE. However, ADE is more challenging to implement in case of unsteady-state flow conditions, especially when lateral exchanges occur. The diffusive wave equation and the advection-diffusion transport equation have very similar mathematical equation, but of course do not describe the same processes. The diffusive wave equation is derived from Saint-Venant continuity and momentum equations and can be applied to a wide range of phenomena in different fields as exposed by Singh (2002) for the kinematic wave, such as flood routing model but also for solute transport (Cimorelli et al., 2014). The advection-diffusion equation is derived from the mass conservation principle applied for matter dissolved in the water and taking into account two basic processes of transport: advection and diffusion in which the Fick's law leading to the diffusive term was applied. Under some hypotheses, the physical

equations of both the diffusive wave equation and the advection-diffusion equation can lead to the same similar mathematical equation which can justify the use of the same resolution approaches. In the present study, following Singh (2002), both phenomena, the diffusive wave equation and the advection-diffusion equation, are treated using the same mathematical approach: the Hayami (1951) analytical solution extended by Moussa (1996) to the case of uniformly lateral flow (and solutes) using an inverse problem approach.

For practical application in karst systems, the knowledge of the temporal distribution of lateral flows and concentrations is to better characterize interactions along the conduit during flood and the hydrogeological functioning of the aquifer. In fact, most karst systems are only accessible punctually, leaving large portions of the system inaccessible for direct observation. Thus, the modelling of the lateral exchanges solving the inverse problem is a tool to decipher the hydrological functioning of such inaccessible conduits located between two monitoring stations. The total volume of water and mass balance of lateral exchanges can be easily estimated by the difference between input and output average values. Thus, the interest of simulating temporal variability is to characterize the evolution of lateral flows and their mineralisation during the flood. This is important because these exchanges can be successively positive or negative during a single flood. The existence or not of a complex dynamic of lateral exchanges cannot be identified without a temporal analysis.

The aim of this paper is to propose a new framework based on a solution of the inverse problem for the advection-diffusion equations (Moussa, 1996) to identify the temporal distribution of lateral flows and their concentrations. We consider the general case where only flow (and eventually solute concentration) is measured on gauging stations without any additional information on lateral flow. The aim is to use simultaneously both measured input and output hydrographs (and eventually input/output solute concentrations) in order to identify lateral inflow/outflow temporal distribution and its concentration. The framework is tested on discharge and total dissolved solids data from two reaches of a karst conduit in the French Jura Mountains, for several flood events under various hydrological conditions. The study case corresponds to the case generally encountered in practice where the spatial variability is unknown. The model simulations - describing the temporal variability of the lateral exchanges - are then used to better characterize the interactions existing along the conduit during floods.

## 2 Modelling approach

### 2.1 Assessing lateral flows

30 **2.1.1 Diffusive wave model without lateral flows**

The diffusive wave equation is an approximation of the Saint-Venant equations used to model 1D unsteady flow in open channels (Moussa and Bocquillon, 1996):

$$\frac{\partial Q}{\partial t} + C_Q \frac{\partial Q}{\partial x} - D_Q \frac{\partial^2 Q}{\partial x^2} = 0 \tag{1}$$

where $x$ [L] is the length unit, $t$ [T] is the time and the celerity $C_Q$ [LT$^{-1}$] and the diffusivity $D_Q$ [L$^2$T$^{-1}$] are functions of the discharge $Q$ [L$^3$T$^{-1}$].

Between two stations $I$ (inflow) and $O$ (outflow), the model is applied on the flood component ($Q_{I,flood}(t)$ and $Q_{O,flood}(t)$) of the total discharge ($Q_{I,tot}(t)$ and $Q_{O,tot}(t)$), which can be deduced by using the inflection point on the hydrograph recession and then removing the baseflow components $Q_{I,base}(t)$ and $Q_{O,base}(t)$:

$$Q_{I,flood}(t) = Q_{I,tot}(t) - Q_{I,base}(t) \tag{2}$$

$$Q_{O,flood}(t) = Q_{O,tot}(t) - Q_{O,base}(t) \tag{3}$$

Eq. (1) is of parabolic type and its resolution requires appropriate additional conditions imposed at the limit of the solution domain ($0 \leq x \leq L$ and $t \geq 0$), $L$ being the channel length. Hayami (1951) assumed the following domain $0 \leq x < \infty$ and $t \geq 0$. The initial condition was for $t = 0$, $Q(x,t) = 0$ for $x \in [0, \infty]$ whereas, two boundary conditions are as follows: for $x = 0$, $Q(x,t) = \delta(t)$ and for $x = X \to \infty$, $Q(x,t) = 0$ where $\delta(t)$ is the Dirac delta function. The resolution of Eq. (1) can

15 either be made using a numerical method or using a convolution. We choose the convolution approach instead of a numerical method for two reasons. First, an analytical solution of the inverse problem for the advection-diffusion equations (Moussa, 1996) is available and easy-to-use. Moreover, the use of a convolution enables to avoid the need for choosing adequate space and time steps in numerical methods (subdividing the reach into space steps $dx$, and the time into time steps $dt$) which may introduce numerical instabilities. Based on the Hayami assumptions (Hayami, 1951), considering $C_Q$ and $D_Q$ as constant

20 parameters over time along a channel network of length $L$, the diffusive wave equation without lateral exchange can be written as follows (Moussa, 1996):

$$Q_{I,routed}(t) = \int_0^p Q_{I,flood}(t-T)K_Q(T)dT = Q_{I,flood}(t) * K_Q(t) \tag{4}$$

where $p$ is the time memory of the system, and the symbol $*$ represents the mathematical convolution function. As there is no problem of time of calculation, the term $p$ must be chosen large in comparison to the travel time on a channel reach, and the

25 time step $dT$ must be chosen very small.

The Hayami kernel function $K_Q(t)$, which is expressed as follows:

$$K_Q(t) = \frac{L}{2(\pi D_Q)^{1/2}} \frac{exp[\frac{C_Q L}{4 D_Q}(2 - \frac{L}{C_Q t} - \frac{C_Q t}{L})]}{(t)^{3/2}} \tag{5}$$

The Eq. (4) is then used to compare $Q_{I,routed}(t)$ to $Q_{O,flood}(t)$ and perform the parametrization of $C_Q$ and $D_Q$ as described afterwards (Section 2.3 & 3.3.2).

### 2.1.2   Diffusive wave model with lateral flows

By considering the existence of lateral flow exchanges along a channel reach, we obtain (Moussa et al, 1996):

$$\frac{\partial Q}{\partial t} + C_Q(\frac{\partial Q}{\partial x} - q) - D_Q(\frac{\partial^2 Q}{\partial x^2} - \frac{\partial q}{\partial x}) = 0 \tag{6}$$

where $q(x,t)$ $[\text{L}^2\text{T}^{-1}]$ is the lateral flow rate per unit length as a function of distance along the channel reach $x$. The expression
$q(x,t)$ may be positive or negative depending on the occurrence of lateral inflow or outflow, respectively (Fig. 1).

In the case of the diffusive wave equation with lateral flows, an analytical resolution is proposed by Moussa (1996) based on the Hayami assumptions, accounting for uniformly distributed lateral flow between two gauging stations $I$ (inflow) and $O$ (outflow).

$$Q_{I,routed}(t) = \Phi_Q(t) + (Q_{I,flood}(t) - \Phi_Q(t)) * K_Q(t) \tag{7}$$

$$\text{with} \quad \Phi_Q(t) = \frac{C_Q}{L} \int_0^t (Q_{A,flood}(\lambda) - Q_{A,flood}(0))d\lambda \tag{8}$$

where $d\lambda$ is the variable along which the integral function is calculated.

Note that Eq. (6) gives the general form of the diffusive wave equation for any spatio-temporal distribution of $q(x,t)$ while Eqs. (7 and 8) give the resolution of Eq. (6) in the particular case of uniformly distribution of $q(x,t)$ along the reach, under the
hypotheses used in the Hayami model ($C$ and $D$ constant).

The lateral flood flows $Q_{A,flood}(t)$ along the reach, under the hypothesis of uniform lateral distribution, is expressed as:

$$Q_{A,flood}(t) = \int_0^L q(x,t)dx \tag{9}$$

The inverse problem enables the identification of the temporal distribution of the lateral inflows or outflows $q(x,t)$ over the channel reach. According to Moussa (1996), by knowing $Q_{I,flood}(t)$ and $Q_{O,flood}(t)$, it is possible to calculate $Q_{A,flood}(t)$.
Eq. 7 gives then:

$$A_Q(t) = Q_{O,flood}(t) - Q_{I,flood}(t) * K_Q(t) \tag{10}$$

$$\text{with} \quad \Phi_Q(t) - \Phi_Q(t) * K_Q(t) = A_Q(t) \tag{11}$$

The resolution of equations (10) and (11) requires first the identification of $K_Q(t)$ using equation (5), and consequently a predetermination of the two parameters $C_Q$ and $D_Q$, to calculate afterwards lateral flow $Q_A(t)$ as follows:

$$Q_{A,tot}(t) = Q_{A,base}(t) + Q_{A,flood}(t) \tag{12}$$

$$\text{with} \quad Q_{A,flood}(t) = \frac{L}{C_Q} \frac{d\Phi_Q}{dt} \tag{13}$$

$$\text{and} \quad Q_{A,base}(t) = Q_{O,base}(t) - Q_{I,base}(t) \tag{14}$$

The Hayami analytical solution of the diffusive wave model assumes a uniformly distributed flow rate $q(x,t)$ along the reach which is the simplest hypothesis when non additional information from the field is available. Moreover, under this hypothesis, and under the hypotheses used in the Hayami model, the unknown $q(x,t)$ is reduced to $q(t)$ in the analytical solution of the inverse problem. Even if the spatial distribution of $q(x,t)$ is unknown, the simulated q(t) under the hypothesis of uniformly spatial distribution will give to the modeller important information on the temporal variability of lateral exchanges, because it enables to distinguish three cases: i) negative $q(t)$ during the whole event; ii) positive $q(t)$ during the whole event; iii) alternating positive and negative $q(t)$ during an event. It enables also to calculate the temporal distribution of lateral flow $q(t)$, and the maximum/minimum values of $q(t)$.

## 2.2 Assessing lateral solute transport

The classical 1D advection-diffusion equation (ADE) for steady-state flow conditions is analogous to the DW equation (Eq. 1), replacing discharge by solute concentration, and celerity and diffusivity parameters by advective velocity and diffusion parameters, respectively. In unsteady-state flow conditions, and when lateral fluxes occur, the application of ADE is not so straightforward. We propose here to assess lateral solute transport (defined by Eq. 15) during flood, applying the DW model accounting for lateral exchanges, as a transfer function. Thus, the analytical solution of Moussa (1996) is used to resolve the conservative solute transport respecting the mass conservation law and accounting for uniformly distributed lateral fluxes.

$$M(t) = S(t).Q(t) \tag{15}$$

where $M(t)$ is the solute flux rate $[\mathrm{MT}^{-1}]$, $S(t)$ is the solute concentration $[\mathrm{ML}^{-3}]$ and $Q(t)$ is the discharge $[\mathrm{L}^3\mathrm{T}^{-1}]$.

As previously described for water flows, the model involves first the determination of the flood ($M_{I,flood}(t)$ and $M_{O,flood}(t)$) and base ($M_{I,base}(t)$ and $M_{O,base}(t)$) components of the total fluxes ($M_{I,tot}(t)$ and $M_{O,tot}(t)$) based on mass-chemograph separation, corresponding to the evolution of solute transport in function of time:

$$M_{I,flood}(t) = M_{I,tot}(t) - M_{I,base}(t) \tag{16}$$

$$M_{O,flood}(t) = M_{O,tot}(t) - M_{O,base}(t) \tag{17}$$

Then, following the method described above for water flows, lateral solute exchange $M_A(t)$ is calculated by adapting Eqs. 5, 7, 10 and 11 as follows:

$$M_{O,flood}(t) = \Phi_M(t) + (M_{I,flood}(t) - \Phi_M(t)) * K_M(t) \tag{18}$$

$$\text{with} \quad \Phi_M(t) = \frac{C_M}{L} \int_0^t (M_{A,flood}(\lambda) - M_{A,flood}(0))d\lambda \tag{19}$$

By analogy with Eq. (5), the kernel function for the solute transport, $K_M(t)$, is expressed as:

$$K_M(t) = \frac{L}{2(\pi D_M)^{1/2}} \frac{exp[\frac{C_M L}{4 D_M}(2 - \frac{L}{C_M t} - \frac{C_M t}{L})]}{(t)^{3/2}} \tag{20}$$

As we use similar mathematical resolution technique for both DWE and ADE, the resolution of the ADE needs two parameters noted the solute flux celerity $C_M$ and the solute flux diffusivity $D_M$ which plays a similar role that the two parameters $C_Q$ and $D_Q$ of the DWE.

Finally, this modelling framework combining the calculation of both lateral water flows $Q_A(t)$ and solute fluxes $M_A(t)$, allows the assessment of the solute concentration of the lateral flows $S_A(t)$ in the case of positive or negative $Q_A(t)$ and $M_A(t)$:

$$S_A(t) = \frac{M_A(t)}{Q_A(t)} \tag{21}$$

## 2.3   Framework

We propose in this section a step by step structure to help readers using our framework to investigate the exchange dynamics of water flows and conservative solute transport along a channel reach between two gauging stations. To simulate lateral exchange flows $Q_A$ and solute fluxes $M_A$ during floods, the required data are discharge and solute time series from both stations covering a complete flood event. The simulated lateral flow and solute fluxes are then used to better characterize the temporal variability of the exchanges occurring during the flood into the karst conduit, giving additional information to better characterize the hydrogeological functioning of the karst aquifer. We assume a priori linearity of both processes: the diffusive wave equation for flow transfer and the advection-diffusion equation for solute transfer. This causes that a superposition is valid (separation of the base flow and the base solute transport can be done) as well as the convolution approach can be applied. Consequently, because of the assumed linearity both problems, unsteady flow and unsteady solute transport, are analysed using a uniform approach because both problems are described using the same type of equation. Therefore the new keys consist that both the diffusive wave equation and the advection-diffusion equation, are treated using the same mathematical approach: the Hayami (1951) analytical solution extended by Moussa (1996) to the case of uniformly lateral flow (and solutes). When the observations at the upstream end and at the downstream end are known then determination of the lateral inflow/outflow constitutes some kind of inverse problem. The problem is solved using the same analytical techniques applied to both the diffusive wave equation and

the advection-diffusion transport equation describing both flow and transport. The modelling of the lateral flood component includes four parameters corresponding to $C_Q$ and $D_Q$ for water flow and $C_M$ and $D_M$ for solute transport. A relationship may exist between the diffusive wave celerity $C_Q$ and the flow velocity $C_M$ in the advection-diffusion equation (for example for rectangular sections, $C_Q = 5/3\ C_M$). Figure 2 gives a graphical representation of this framework whose 7 stages are listed below.

(1) Collection of discharge ($Q_{I,tot}$ & $Q_{O,tot}$) and concentration ($S_{I,tot}$ & $S_{O,tot}$) data from input and output stations.

(2) Calculation of the total solute fluxes $M_{I,tot}$ and $M_{O,tot}$ using Eq. (15).

(3) Determination of the base and flood components from the hydrograph ($Q_{I,tot}$, $Q_{O,tot}$) separation defined by Eqs. (2) & (3) and mass-chemograph ($M_{I,tot}$ and $M_{O,tot}$) separation defined by Eqs (16) & (17). The base and flood components are separated with the constant slope method (McCuen, 2004) by using the inflection point on the hydrograph recession. The inflection point determined on the hydrograph is used to separate base and flood components for both hydrograph and mass-chemograph.

(4) Calculation of the lateral base component for water flow ($Q_{A,base}(t)$) and solute fluxes ($M_{A,base}(t)$), using Eq. (14).

(5) Modelling of the lateral flood exchanges using two steps.
A. First, the Eq. (4) to parametrize $C_Q$ & $D_Q$ and $C_M$ & $D_M$ with a trial-and-error optimization, calculating $Q_{I,routed}$ and $M_{I,routed}$ to get the best fit with $Q_{O,flood}$ and $M_{O,flood}$, respectively. The model is parsimonious with only the two parameters (celerity and diffusivity) to optimize. These two parameters are independent, hence the parametrization procedure using a trial-and-error optimization give similar results as an automatic optimization with a quick convergence to a unique and a stable solution. Nevertheless, if the user needs to optimize more parameters, an automatic optimization procedure is necessary.
B. Afterwards, the inverse problem is applied to simulate lateral flood exchanges $Q_{A,flood}(t)$ and $M_{A,flood}(t)$, using Eqs. (10), (11) & (13).

(6) Simulation of the total lateral exchanges $Q_{A,tot}(t)$ and $M_{A,tot}(t)$, using Eq. (12).

(7) Calculation using Eq. (21) of the solute concentrations of lateral base and flood flows $S_{A,base}(t)$ and $S_{A,flood}(t)$, respectively. The determination (if possible) of the total solute concentration $S_{A,tot}(t)$ follows then.

The proposed framework is generic enough to explore saturated and unsaturated conditions, base flow and floods, water and suspended particulate matter or any other tracer concerned by the advection-diffusion equation, considering the analogy with the diffusive wave equation. Moreover, the numerical resolution opted for leans on the analytical resolution of the diffusive wave taking into account uniformly distributed lateral flows.

In our study, the application of this framework is done separately on various selected flood events. Hence, the two parameters set ($C_Q$ & $D_Q$ and $C_M$ & $D_M$) are calibrated for each event. Then, the relationships between the parameters and the variations of water flow and solute transport are analysed. It can be expected for example that $C_Q$ & $D_Q$ should increase with more pronounced flood peaks. Furthermore, the model simulations performed along the reaches allow estimating the temporal variability of the lateral exchanges. The simulations can hardly be compared to punctual field measurements, however they are used as a diagnostic tool to better characterize the exchange dynamics occurring along the conduit during flood. The compilation of all results leads us to define a functional scheme of the studied karst system.

Note that if additional information on lateral fluxes is available, as for example punctual inputs/outputs on the reach, information on the amplitude of the spatial distribution of lateral flows, or measurements of solute concentrations on different points, the framework proposed herein is generic and can be easily used. In this case, the studied zone has to be subdivided into different reaches with eventually punctual inputs/outputs on some nodes; then the inverse problem can be applied on each reach. Moreover, if additional variables are measured, as for example piezometer levels or hydrographs on tributaries (or concentrations in the water table or tributaries), a validation can be undertaken by comparing the measured variable to the simulated lateral flow hydrograph. But the gain in understanding the complexity of the studied karstic systems is worth the relative loss of "lateral" precision, that is, soon as the tributaries are not the key concern, i.e. the main reaches can be properly identified.

## 3   Study site

### 3.1   Field situation

The study site is located in the Doubs river valley at the northern limit of the Jura Mountains in Eastern France, near the village of Fourbanne ($47°19'54"N6°18'15"E$, Fig 3a). The Fourbanne site is one of the experimental sites of the "Jurassic Karst" hydrogeological observatory ($http://zaaj.univ-fcomte.fr/spip.php?article13$) and monitored continuously since December 2013. The local geological structure is characterized by tabular Jurassic limestones and shales, crosscut by N-S trending normal faults (Charmoille, 2005). The recharge area of the site covers about 30 km$^2$. The site was selected for its dominantly allogenic recharge and its well-developed, partially accessible conduit network.

The upstream recharge area corresponds to a surface watershed underlain by impervious lower Jurassic marls. The conduit network is fed by swallow holes located at normal faults, bringing into contact lower Jurassic shales and karstified middle Jurassic limestones (Fig. 3b). The Verne swallow hole constitutes the main infiltration point of the Fourbanne karst system and corresponds to monitoring station s1. The allochtonous recharge joins the well-developed cave stream of the "En-Versennes" cave, which is explored over a distance of about 8 km. An artificial well near the village of Fontenotte gives direct access to the cave stream, where monitoring station s2 was installed, about 5 km downstream of station s1. The Fontenotte cave stream can be followed for another 2 km downstream of station s2, where it disappears into an inaccessible conduit network. It joins finally the Fourbanne spring, which is fed by a saturated siphon of 25 m depth and explored by cave-divers over the last 500

m. Monitoring station s3 was installed at the Fourbanne spring.

The climate is temperate with both oceanic and mountainous influence. Rainfall averages 1200 $\mathrm{mm/year}$, occurring mainly in autumn and winter, but with slightly higher intensities in summer (Vermot-Desroches, 2015). Based on long-term records in the Jura Mountains, the number of rainy days per year is 140 on average (Charlier et al., 2012), corresponding for the most part to low-intensity rainfall events: 50% of rainy days had less than 3 mm of rainfall, whereas days between 15 and 30 $\mathrm{mm}$ of

rainfall represented only 10% of rainy days.

### 3.2 Field monitoring and data processing

Discharge ($Q$) and electrical conductivity ($EC$) were monitored continuously at 15-minute intervals from January to June 2015 at the three stations (Fig. 3b): the Verne swallow hole (station s1), the Fontenotte cave stream (station s2,) and the Fourbanne spring (station s3). Stations s1 and s2 were equipped with OTT CTD probes for conductivity, temperature, and water level monitoring, whereas a OTT-Hydrolab DS5X multiparameter probe was used at station s3 for conductivity and water temper-

5  ature, and a OTT Orpheus mini probe for water level. The three stations divided the main conduit into two reaches: R1 from s1 to s2 (3.1 $\mathrm{km}$) and R2 from s2 to s3 (5.4 $\mathrm{km}$, Fig 3c). Hourly precipitation were recorded by a fully automatic Campbell BWS200 weather station installed next to monitoring station s2 in the Fontenotte village.

From the available time series, 7 flood events with complete data sets for all stations and various rainfall intensities were

10  selected. A synthetic characterization of the 7 events is given in Fig. 4, where they are sorted from event 1 to 7 in function of decreasing baseflow at system outlet (station s3). The event duration varied from 24 to 52 hours with total precipitation amounts between 3.4 to 46.6 $\mathrm{mm}$. A progressive decrease of the hydrogeological response with decreasing baseflow was found for events 1 to 5, whereas events 6 and 7 from an extremely dry period in June 2015 behaved differently with an important inflow from the station s1.

The EC is directly related to the Total Dissolved Solids (TDS) assuming that TDS represent mainly conductive ionic compounds. The TDS values were therefore calculated directly from EC by using a constant factor of 0.64 (100 $\mathrm{S.cm^{-1}}$ = 64 $\mathrm{g.l^{-1}}$), which is commonly used by OTT CTD probes and consistent with the literature (Lloyd and Heathcote, 1985) according to the water mineralization range of the data set.

### 3.3 Model application to the study site

To illustrate the model behaviour described theoretically in Section 2 and to help to define a parametrization strategy, this section presents a sensitivity analysis on a benchmark flood event. This event was defined in order to have similar characteristic

(same range of magnitude and parameter's values) than those presented in the model application.

### 3.3.1 Sensitivity analyses

This analysis was carried out on the celerity $C_Q$ and the diffusivity $D_Q$ of the modelling approach both applied on water flows. The celerity $C_Q$ and the diffusivity $D_Q$ describe the propagation and the flattening of the flood peak, respectively. The model applied on solute fluxes using $C_M$ and $D_M$ play similar roles as $C_Q$ and $D_Q$.

Figs. 5a-a' represent the routed input hydrograph and the simulated lateral flow, respectively, varying $C_Q$ from 0.1 to 0.3 $\mathrm{m.s}^{-1}$ with a fixed $D_Q$ of 0.1 $\mathrm{m}^2.\mathrm{s}^{-1}$ (i.e. low $D_Q$ value). Similarly, Figs. 5b-b' represent the same graphs, but with a fixed $C_Q$ of 0.15 $\mathrm{m.s}^{-1}$ (i.e. low $C_Q$ value) and varying $D_Q$ from 0.1 to 10 $\mathrm{m}^2.\mathrm{s}^{-1}$.

Figs. 5a-b illustrate the application of the Eq. (4) simulating the propagation of the input signal in order to fit the output signal with the two parameters set ($C_Q$ & $D_Q$) by considering no lateral exchange. The graphs clearly demonstrate that the routed input signal is much more sensitive to celerity than diffusivity. Varying $C_Q$ by a factor of 3 has a stronger impact on the results

than varying $D_Q$ by a factor of 100. As an example, a 10% variation of $C_Q$ (with constant $D_Q$) modifies the maximum of $Q_{I,flood} * K_Q$ of 9%, while 10% variation of $D_Q$ (with constant $C_Q$) changes to vary the maximum of $Q_{I,flood} * K_Q$ of 0.6% only. Moreover, this sensitivity test illustrates the impact of both parameters on the propagation velocity and the shape of the flood peaks: increasing $C_Q$ values yield more rapidly propagating and higher flood peaks, whereas increasing $D_Q$ values lead to flattened peaks. These observations are in agreement with the literature (Moussa and Bocquillon, 1996; Yu et al., 2000;

Chahinian et al., 2006; Charlier et al., 2009) and confirm that the lower $C_Q$ and the higher $D_Q$, the lower the peak flow intensity and the transfer velocity.

Figs. 5a'-b' illustrate the simulation of lateral flows using the solution of the inverse problem knowing input and output signals. It shows that $C_Q$ and $D_Q$ influence the lateral flow rates as well as the exchange direction (inflows or outflows) which could be reversed during a same flood event. Low $C_Q$ and $D_Q$ values lead to lateral inflows followed by outflows, whereas high $C_Q$

and $D_Q$ values lead to outflows followed by inflows.

### 3.3.2 Parametrization strategy

From the sensitivity analysis, a parametrization strategy was defined for the application of the Eq. (4) for the four parameters: $C_Q$ and $D_Q$ for discharge, and $C_M$ and $D_M$ for solutes flux. The parameters were optimized by the trial-and-error method

based on simulations of the routed input signals and by using the following criteria: i) C was determined first from the peak delay between two succeeding gauging stations (celerity = reach length / delay), ii), whereas D was adjusted in order to get a best fit of the shape of the hydrograph or mass-chemograph with the observed output signal.

# 4 Results

## 4.1 Simulation of lateral exchanges

Figure 6 illustrates the application of the framework (depicted in Fig. 2) for reaches R1 and R2 of the study site for flood event no. 1 with 21 mm of rainfall during high flow conditions.

The lateral base exchanges calculated for reach R1 demonstrate that the output signal observed at station s2 for discharge and solutes cannot be entirely explained by the contribution of the input signal, but that it was due lateral inflows between s1 and s2. The solute transport model shows in addition that the lateral inflows were strongly mineralized, with higher TDS values than for stations s1 and s2. The solute fluxes of the base component at station S2 were thus essentially derived from lateral inflows along reach R1. This pattern was different for reach R2, where lateral base inflows were half these of and with similar TDS values.

Regarding the flood components, the simulated lateral exchange indicates important lateral inflow and high solute influx along R1. The concentration estimations indicate a similar evolution than for station s2, which is characterized by a TDS dilution during the flood. In contrast, dynamics were totally different along R2, where outflows were simulated. The concentration estimations of the lateral component deduced from outflows and solute outfluxes are very low compared to the measurements at stations s2 and s3, suggesting the presence of more complex processes than simple outflows, as will be discussed later on in Section 5.2.

## 4.2 Variability of lateral exchanges

Following the example detailed for flood event no. 1 in the previous section, this section aims at summarizing results of the model application on all selected flood events in order to get informations on the general hydrological functioning of the field site.

### 4.2.1 Lateral exchanges for the base component

Figures 7a-c present lateral exchanges for base water flow and base solute fluxes for all events of reaches R1 (orange labels) and R2 (purple labels). When comparing the modelled mean lateral base flow exchange in function of the measured mean base input, two distinct linear relationships can be observed (Fig. 7a). For both reaches, lateral exchanges were positive, indicating inflows that increased linearly with average base input. For reach R1, lateral water inflow was 2.6 times higher than mean input flow from station s1, whereas for reach R2 the mean lateral water inflow represented only 0.4 times the mean inflow from station s2. For solute transport (Fig. 7c) a very similar relationship was found. However, for R1 the slope of the correlation was much steeper than for water flow (4.3 against 2.6), indicating that the lateral inflow water was more mineralized than the input

flow from station s1 already present in the system. In contrast, for reach R2 a slightly lower slope was found for solutes than for water flow (0.3 against 0.4), meaning that lateral inflow was probably a little less mineralized than input flow from station s2.

### 4.2.2  Lateral exchanges for the flood component

Figs. 7b-d illustrate lateral water flows and solute transport calculated for flood flow. To summarize the dynamics of the lateral
exchanges during the flood, minimum and maximum values are presented rather than average values in order to characterize intensities of both lateral gains and losses which may occur during the same event. Along reach R1, two distinct groups are observed depending on peakflow from input station s1 (Fig. 7b): (i) Events 1 to 5 with low input values are characterized by lateral inflows, with high maxima and minima close to zero. (ii) Events 6 and 7 with high input values from strong rain events during an extremely dry period, show maxima close to zero and strongly negative minima indicating important lateral losses.
Comparing the slopes of the linear relationships for water flows (Fig. 7b) and solutes flux (Fig. 7d), it appears that the water inputs of the first group were strongly mineralized, whereas the water losses of the second group were characterized by low mineralization. For reach R2, all maxima of water flow and solutes flux are close to zero, whereas the minima are negatively correlated with input values, indicating increasing lateral losses with increasing peakflow. Comparing the slopes between water flood flow (fig 7b) and solute flood flux (fig 7d), it appears that lateral losses were systematically less mineralized than input
water from station s2.

## 4.3  Transport dynamics along the conduit network

### 4.3.1  Distribution of model parameters

In this Section we present the distribution of values for the model parameters - celerity and diffusivity - in function of maximum
input flood flow intensities, with the aim to retrieve information on flow and transport dynamics (Fig. 8).

The $C_Q$ water flow celerity parameter increased linearly with input peakflow for events 1 to 5 for reaches R1 and R2 (Fig. 8a, graph in semi-log scale). Reach R1, entirely located in the unsaturated zone, showed lower celerities compared to reach R2, which span both the unsaturated and saturated zone. Events 6 and 7, corresponding to the extreme dry period, did not fit this
trend for both reaches, and were characterized by lower $C_Q$ values. This behaviour may be related to the low degree of water saturation of the system at the beginning of the flood. The $C_M$ solute flux celerity parameter showed similarly a relationship with the flood flow intensities for events 1 to 5, but only for reach R1 and again with a different behaviour for events 6 and 7 (Fig 8c). On the contrary, we did not find any relationships for reach R2, suggesting a more complex behaviour for this part of the system, most probably related to the presence of the saturated zone.

All events from reaches R1 and R2 had very low $D_Q$ diffusivities. Only events 1-4 on reach R2 showed increasing $D_Q$ values with increasing input peakflow (Fig 8b). We note that these 4 data points were all from a wet period and from reach R2, which localized in the non-saturated and the saturated zone. The $D_M$ solute flux diffusivity parameter showed a very similar distribution as $D_Q$, i.e. again with high values only for events 1-4 on reach R2 (Fig 8d). Finally, note that a relationship may exist

between the diffusive wave celerity $C_Q$ and the flow velocity CM in the advection-diffusion equation such as $C_Q \cong 5/3 C_M$ in the case of a rectangular section. Fig. 8 shows that the ratio $C_Q / C_M$ ranges between 1.1 and 2.3.

### 4.3.2    Assessment of the saturated level of the conduit network

The analysis of the parameter distributions showed distinct trends for R1 and R2, which can be attributed to the presence of the saturated zone in the lower parts of reach R2 (see Figure 3 for the hydrogeological scheme of the main conduit). In fact, flood routing in the unsaturated zone is related to flow in open conduits, whereas in the saturated zone it is controlled by pressure

transfer leading to an almost instantaneous propagation.

Consequently, the difference of flood routing between unsaturated and saturated conduits along the system should be observed in the $C_Q$ parametrization – characterizing the propagation of the flood peak along the 2 reaches R1 and R2. In fact, by considering a mean constant slope of the main karst conduit, we hypothesize a constant $C_Q$ value along the unsaturated zone in R1

and R2. The relation between $C_Q$ values of R1 and R2 for each event should therefore deliver information on the localization of the limit between the saturated and the unsaturated zone. Based on these hypotheses, the Eq.(22) is used to estimate the percentage of conduit length of reach R2 located within the unsaturated zone, defined as "U" value.

$$U = \frac{C_Q(R1)}{C_Q(R2)} \cdot 100 \qquad (22)$$

This limit is supposed to fluctuate in function of the baseflow condition. Figure 9a represents the U values of all events in function of the mean baseflow at the system outlet in station s3 (Fourbanne spring). A linear relationship is observed for events 1 to 6 showing - as expected - an increasing U with a decreasing mean baseflow, which is coherent with the saturation level of the system measured by discharge. Based on the calculated U value and with an estimated total length of 5.4 km for R2, the saturated conduit length is approximatively 1.5 to 2.2 km. Event 7 shows that the system behaved differently for flood events

during extreme dry periods. This point will be discussed in the next Section.

# 5 Discussion

## 5.1 Modelling framework

Our study intends to present a new framework to quantify the temporal evolution of lateral flows and their concentrations during floods in a well-developed karst conduit networks. It uses the diffusive wave (DW) model, which is a physically-based approach, parsimonious and easy-to-use. The inverse problem was used to simulate both lateral flows and solutes flux under unsteady-state conditions following the assumption of uniformly distributed lateral exchanges. Our modelling approach is not used with data sets allowing a validation of the computed lateral fluxes. In the study case, there is no monitoring of the tributaries/losses along the 2 reaches of the study site. However, if additional variables are measured, as for example piezometer levels or hydrographs on tributaries, a validation can be undertaken by comparing the measured variable to the simulated lateral flow hydrograph. It has been proposed by Charlier et al. (2015), in which the hydrograph dynamics of lateral springs is compared with the simulated lateral exchanges. Regarding parametrization, the analysis of the distribution of water flow parameters (celerity $C_Q$ and diffusivity $D_Q$) of 7 flood events allowed to characterize the variability of flood routing in more detail for the reach R1 located in the unsaturated zone and the reach R2 covering both unsaturated and saturated zones. Both reaches showed linearly increasing $C_Q$ values with increasing flood intensities (Section 4.3.1.). The difference of $C_Q$ parametrization between the two reaches was then used in Section 4.3.2 to quantify the fluctuation of the unsaturated/saturated boundary within the karst conduit. solute transport parameters ($C_M$ and $D_M$) are less obvious to interpret. Reach R1 showed a linear relationship of $C_M$ in function of solute flux intensity, whereas a comparable trend was absent for reach R2 attesting the difficulty to characterize transport processes within the saturated conduit network from the available monitoring design.

The proposed framework requires to decompose the base and the flood components, being consistent with the duality of flow processes dynamics. As proposed by many authors (Atkinson, 1977, among others), the base component typifies slow flow in the system, which is strongly influenced by interaction with the rock matrix (or low permeable volumes) with high storage capacities, whereas the flood component typifies quick flow within the conduit network with low storage capacities, which is strongly linked with flood intensity. Thus, considering that baseflow is used to estimate base exchanges and the diffusive wave flood routing model to assess flood exchanges, our framework gives a deconvolution of the two components of the lateral exchanges helping to interpret the involved processes.

Our methodology quantifies total lateral water flow and solute flux exchange for a given reach, but does not allow to identify simultaneously occurring local lateral flows and fluxes as already highlighted by several authors (Payn et al., 2009; Szeftel et al., 2011; Charlier et al., 2015). The results for the reach R2 furthermore denote the difficulty to decompose exchanges that occurred in part in the unsaturated and in part in the saturated zone. Lateral flood outflows for R2 were mainly combined with low flood outfluxes with total mineralisations lower than those observed for the input and output stations. This may be the result of complex exchanges occurring between conduit and matrix (or low permeable volume) water within the saturated zone, with out- and inflows occurring concomitantly during the same flood and having contrasted concentrations. The DW model

furnishes thus numerous information about lateral exchanges in karst aquifers. However, as reported by Szeftel et al. (2011), we point out the limits of our framework to identify a single, unambiguous model structure representing transport processes when boundary conditions are poorly described. Its limitation is that it considers a reach of a conduit system as a "homogeneous" unit with uniformly distributed lateral exchanges. It is a diagnostic tool which naturally cannot be used to decipher too much complexity in the case of too remote monitoring stations with highly variable lateral contributions.

An interesting point would be to discretize the conduit network in smaller reaches. Nevertheless, it should require a large set of monitoring stations that are not always accessible in karst systems. In this way, Moussa (1997) proposed a methodology to identify the transfer function based on the Hayami analytical solution using the topological elevations of a distributed channel network flood routing modelling. However, in karst conduit system, besides the partial access to the conduit, the karst conduit elevations may not always be relevant to describe flood routing propagation, specifically in the saturated zone.

## 5.2    Functional scheme

The modelling framework is proposed as a diagnostic tool to assess the dynamics of lateral exchanges between the main conduit and the neighbouring compartments of the karst aquifers during floods. The selected events are typified by various initial

baseflow condition and flood flow intensities allowing thus to characterize both low flow and high flow periods providing a rather generic of the hydrological behavior of the system. The model application on several contrasted events on the Fourbanne karst network gives several points of discussions to evaluate exchanges in various hydrological conditions and allow to dress a general hydrological functional scheme of the site. This scheme, presented in Fig. 10, highlights the lateral contributions for the base and the flood components. In fact, we assume that - as discussed before - base and flood components are mainly

related to specific processes as conduit/matrix exchanges distinguished by slow and fast flow, respectively.

(a)  For major precipitations during high flow periods (events no. 1 and 2; Fig. 10a) we showed the importance of lateral exchanges for both base and flood components. For the baseflow component both reaches were fed by higher mineralised lateral inflows. However, for the flood component the model yielded different results for the two reaches. Reach R1,

located entirely in the unsaturated zone, had lateral inflows which were more mineralized than the sinking stream at the reach input. We relate these mineralized inputs to the arrival of tributaries from adjacent sinkholes (see their location on Fig. 3). On the contrary, reach R2 shows less mineralized base inflows and mainly slightly mineralized flood outflows, indicating high losses, which we relate to mixing processes in the saturated part of the conduit.

(b)  For minor precipitations during low flow periods (events no. 3, 4 and 5; Fig. 10b) reach R1 showed similarly mineralized inflows from the base and the flood components, but with a lower inflow amount in coherence with the lowest rainfall intensity of these events. On the contrary, the baseflow component of reach R2 was mainly characterized by inflows, whereas weak outflows were found for the flood component. This shows that the system was mainly influenced by the

drainage of water from the rock matrix even during low flow periods. The weak outflows found for the flood component

of reach R2 probably occurred within the saturated zone.

(c) For major precipitations during extremely dry periods (events no. 6 and 7; Fig. 10c) a quite different behaviour was observed. Reach R1 presented very low base inflows with similar mineralizations as for the input flow, probably from sinking stream tributaries joining the unsaturated conduit network. However, important outflows were observed for the

flood component, indicating high losses towards the rock matrix in the unsaturated zone. On the contrary, reach R2 presented very low baseflow inputs of highly mineralized water. The flood component was characterized mainly by lateral outflows followed by lowest inflows . It is interesting to denote this reversal of the lateral flood flows from out- to inflows, indicating an evolution of the exchanges during the flood event, that we may be related to conduit/matrix interactions in the saturated zone.

Even if we observed increasing lateral inflows with increasing baseflow conditions, a slow inflow component remained always present all along the network (i.e. in the unsaturated as well as in the saturated zone), whatever the hydrological conditions (cf. section 4.2.1.). These constant inflows could probably originate mainly from diffuse infiltration in the unsaturated zone and

from lateral drainage systems in the saturated zone. However, the mineralization of the infiltrating water was higher along R1 compared to R2, reflecting probably different recharge mechanisms. It seems that R1 collected additional inflows from strongly mineralized secondary tributaries mainly localized in the upstream part of the aquifer (R1), in accordance with the presence of sinkholes in the north-eastern part of the study area (Fig. 3).

In Section 4.2.2., from the analysis of the event's distribution according to rainfall intensity and initial baseflow we could distinguish 2 distinct groups of events depending on the general hydrological context: (1) events 1 to 5 during periods of high and low flow, and (2) events 6 & 7 from an extremely dry period. The evolution of the lateral flood exchange for group (1) increased linearly with flood intensity (maximum of the peak flood, Fig. 7), suggesting that these lateral inflows, probably derived from secondary conduits, were proportional to the discharge measured at the input station. For reach R1, the mineralisation of these

inflows increased with maximum input inflow, whereas for reach R2 the mineralisation of outflows decreased. For group (2), lateral flood outflows were observed along both reaches, and notably for R1, meaning that the inflows from secondary lateral conduits observed for group (1) stopped during extreme dry periods in the unsaturated zone. This change of behaviour of the distinct groups of events showed the non-linearity of the lateral exchanges depending on hydrological conditions. This threshold effect in the hydrogeological response may be related to the presence of epiphreatic conduits (Jeannin, 2001) leading to

time-variant limits of their recharge area (Jukić and Denić-Jukić, 2009; Charlier et al., 2012).

To our opinion, from the available data, two types of outflows along the karst conduit network can be described in our conceptual model. The first type corresponds to outflows observed along R1 during the extremely dry period. These outflows occurred

in the unsaturated zone of the conduit at extremely low baseflow conditions where only very few base inflows were observed. Thus, during this period, the flood input following major precipitations recharged the low permeability volume (or matrix) from the conduit network. The second type corresponds to outflows observed along R2 for all baseflow conditions. These outflows seemed to occur in the saturated zone and are related to flow reversal occurring in the saturated conduit as mentioned by several authors (Jeannin, 1996; Maréchal et al., 2008; Bailly-Comte et al., 2010; Binet et al., 2017) who demonstrated the influence of the contrasted kinetics fluctuations of the hydraulic head between the conduit and the surrounding matrix during a flood event. This conduit/matrix relationship in the saturated zone probably induces a complex water mixing effect inside of karst conduit during the flood event.

Besides these results, the flood routing parameters are also indicators for hydraulic processes in the conduit. They are used in section 4.3.2. to estimate the fluctuation of the limit between the unsaturated and the saturated zone within reach R2. We evaluated that the saturated zone occupied 25% to 40% of R2, depending on flow conditions, corresponding to the 1.5 to 2.2 km of conduit next to the outlet station s3. The Fourbanne karst system with its total length of 8.5 km (3.1 km for R1 and 5.4 km for R2) is thus a shallow-phreatic aquifer. Consequently, we demonstrated that besides the impact of the saturated zone on the matrix/conduit exchanges, the unsaturated zone plays a major role on the flood genesis in karst aquifers. Our results confirm that the unsaturated zone has essentially a transfer function – as it is commonly conceptualized (Bakalowicz, 2005) – leading to a rapid hydrological response at the spring due to the high connectivity of the conduit networks. But in our case, we denoted also that a large part of water storage (assessed by the baseflow component in the different stations) originate from this zone. This result highlighted the important storage function of the unsaturated zone as previously pointed in works using hydrochemical approaches (Emblanch et al., 2003; Mudarra and Andreo, 2011), but often minimized in current conceptual karst models.

# 6   Conclusion

The study aims to propose a framework to characterize the spatio-temporal variability of lateral exchanges for flows and fluxes in a karst conduit network, known to have large amount of concomitant in- and outflows during flood events. The main interest in our study is treating both phenomena, the diffusive wave equation and the advection-diffusion equation, with the same mathematical approach assuming uniform lateral flow and solutes, solving the inverse problem of the advection-diffusion equations using an analytical solution. In fact, as the model was applied for two different variables, the flow and the solute transport, a crossed analysis has been performed in order to characterize a functioning scheme of the studied karst system. We showed various lateral exchanges between both unsaturated and saturated zone, we estimated the fluctuation limit of the saturated zone in the conduit and we illustrated the non-linearity of the hydrogeological response related to the initial hydrological conditions.

One of the main points was the ability of our approach to propose a deconvolution of the output hydrograph as well as mass-chemograph allowing to quantify the lateral contributions in terms of flows and mineralization. It was useful to identify water origin of lateral flows and make hypothesis on the flood generation in karst aquifers. The modelling approach uses all data available on the reach in both input/output time series, leading to use our framework as a diagnostic tool to help decompose time series and investigate more precisely lateral exchanges. The results showed that this diagnostic step give new keys to investigate the hydrogeological functioning of karst aquifer and demonstrated for instance that further hydrogeological model development for the case study has to take into account storage in the unsaturated zone and matrix-drain relationships during floods.

*Acknowledgements.* The authors wish to thank Bruno Régent for his active contribution on the field. Many thanks to the Speleology Association of Doubs Central (ASDC) for the precious help on the field and their support to access to the Fontenotte river cave stream in the En-Versennes karst network. We thank Jacques Prost for the welcoming of our monitoring equipment and to give us access to the Fourbanne spring. We also thank the Verne municipality to let us monitoring the Verne swallow hole. The Jurassic Karst hydrogeological observatory is part of the INSU/CNRS national observatory of karstic aquifers, SNO KARST ($http://www.sokarst.org/$). The authors are very grateful to the Editor Mauro Giudici and the four reviewers for their constructive comments of the manuscript. This work was carried out with the financial support of the Burgundi-Franche-Comté Region and the BRGM.

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

## Appendix A: Notations

**Table A1.** Notations

| Symbols | Units | Definitions |
|---|---|---|
| $*$ | $-$ | represents the mathematical convolution function |
| $C_M$ | $[\mathrm{m.s^{-1}}]$ | parameter controlling the celerity of solute flux |
| $C_Q$ | $[\mathrm{m.s^{-1}}]$ | flood wave celerity |
| $D_M$ | $[\mathrm{m^2.s^{-1}}]$ | parameter controlling the diffusivity of solute flux |
| $D_Q$ | $[\mathrm{m^2.s^{-1}}]$ flood wave diffusivity | |
| $(K_Q, K_M)$ | $-$ | Hayami kernel function for water flow and solute transport modelling, respectively |
| $m$ | $[\mathrm{kg.s^{-1}.m^{-1}}]$ | lateral solute flux per length unit |
| $M$ | $[\mathrm{kg.s^{-1}}]$ | solute flux |
| $(M_{I,base}, M_{I,flood}, M_{I,tot})$ | $[\mathrm{kg.s^{-1}}]$ | upstream base, flood and total solute flux, respectively |
| $(M_{O,base}, M_{O,flood}, M_{O,tot})$ | $[\mathrm{kg.s^{-1}}]$ | downstream base, flood and total solute flux, respectively |
| $(M_{A,base}, M_{A,flood}, M_{A,tot})$ | $[\mathrm{kg.s^{-1}}]$ | lateral base, flood and total solute flux exchanges, respectively |
| $q$ | $[\mathrm{m^2.s^{-1}}]$ | lateral flow per length unit |
| $Q$ | $[\mathrm{m^3.s^{-1}}]$ | discharge |
| $(Q_{I,base}, Q_{I,flood}, Q_{I,tot})$ | $[\mathrm{m^3.s^{-1}}]$ | upstream base, flood and total flow, respectively |
| $(Q_{O,base}, Q_{O,flood}, Q_{O,tot})$ | $[\mathrm{m^3.s^{-1}}]$ | downstream base, flood and total flow, respectively |
| $(Q_{A,base}, Q_{A,flood}, Q_{A,tot})$ | $[\mathrm{m^3.s^{-1}}]$ | lateral base, flood and total flow exchanges, respectively |
| $(R1, R2)$ | $-$ | reach 1 (s1 to s2) and reach 2 (s2 to s3), respectively |
| $(s1, s2, s3)$ | $-$ | monitoring station 1, 2 and 3, respectively |
| $(S_{I,base}, S_{I,flood}, S_{I,tot})$ | $[\mathrm{g.l^{-1}}]$ | upstream solute base, flood and total concentrations, respectively |
| $(S_{O,base}, S_{O,flood}, S_{O,tot})$ | $[\mathrm{g.l^{-1}}]$ | downstream solute base, flood and total concentrations, respectively |
| $(S_{A,base}, S_{A,flood}, S_{A,tot})$ | $[\mathrm{g.l^{-1}}]$ | lateral solute base, flood and total concentrations, respectively |
| $t$ | $[\mathrm{s}]$ | time |
| $U$ | $-$ | percentage of the unsaturated conduit length along R2 |
| $x$ | $[\mathrm{m}]$ | downstream distance |

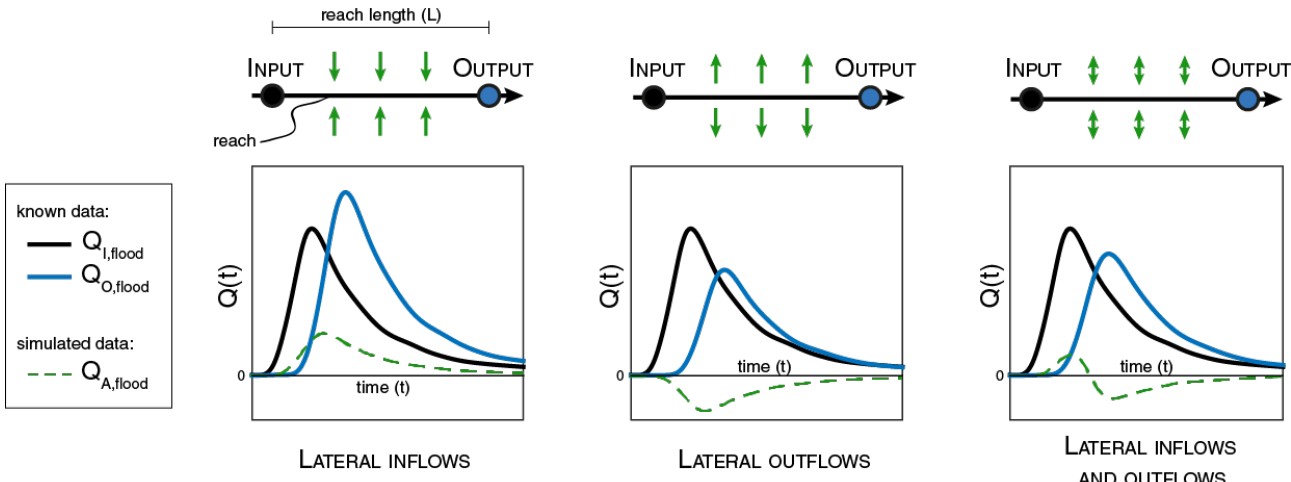

**Figure 1.** Diffusive wave equation to model lateral flood flow exchanges along a channel reach of length $L$. The black curve $Q_{I,flood}$ depicts the evolution of the flow rate with time at the beginning of the channel (input), and the blue curve $Q_{O,flood}$ at the end (output). The dashed green curve $Q_{A,flood}$ corresponds to the lateral flow exchanges which are positive for lateral inflows or negative for lateral outflows.

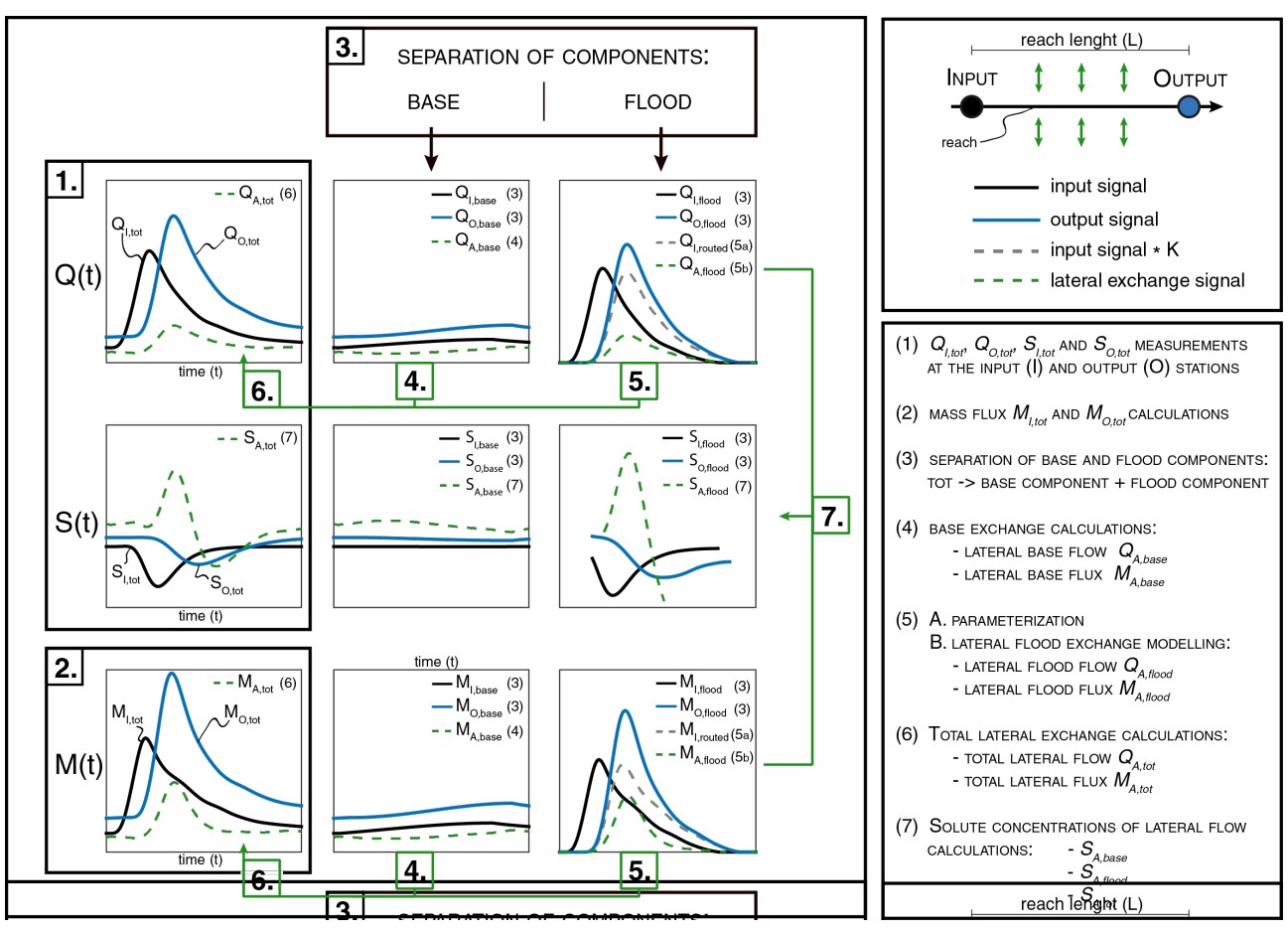

**Figure 2.** Framework to investigate lateral exchange dynamics of water flows and solute fluxes along a channel reach.

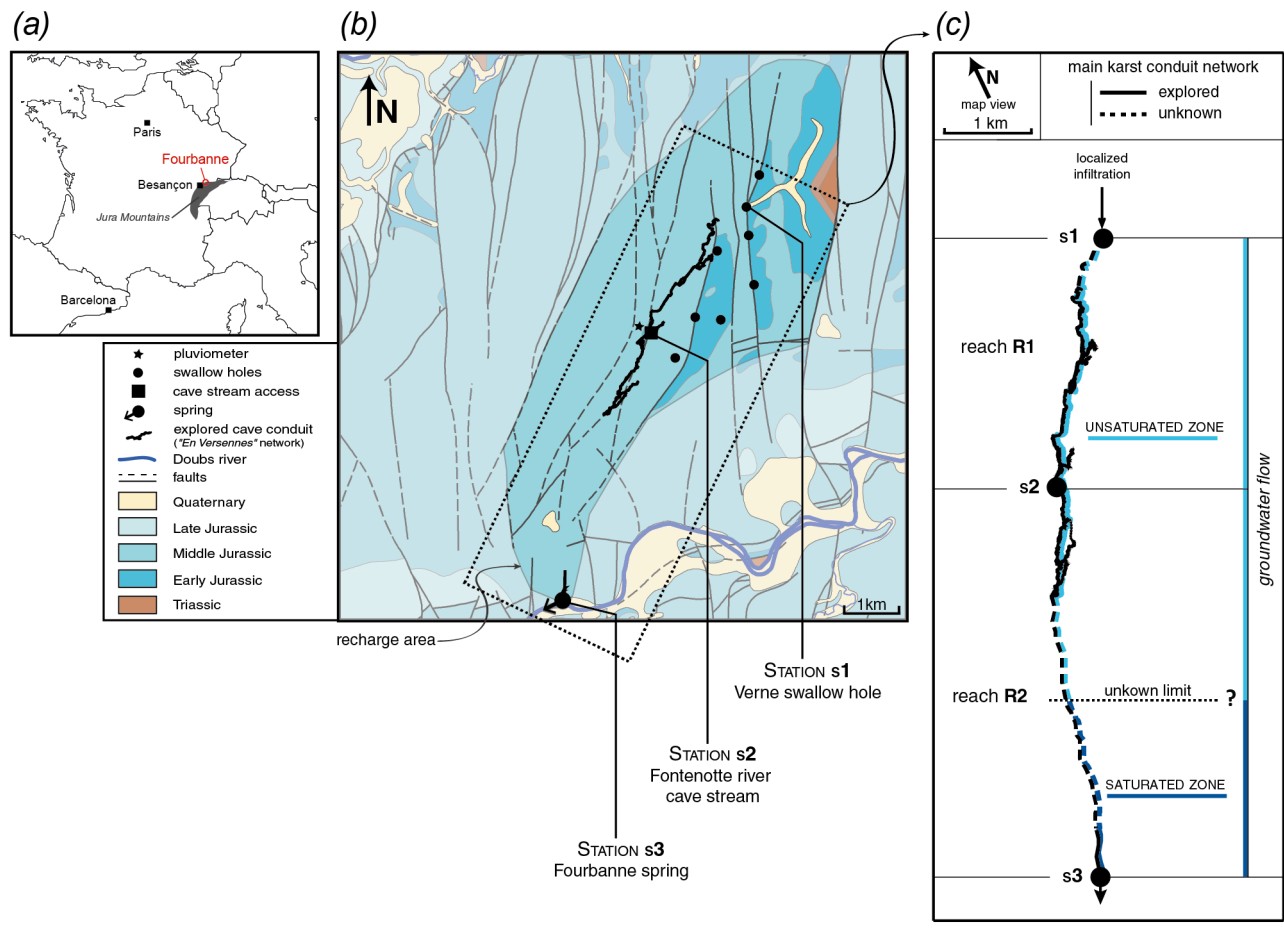

**Figure 3.** The Fourbanne karst system: (a) geographical localization, (b) hydrogeological map, (c) scheme of the main karst conduit network.

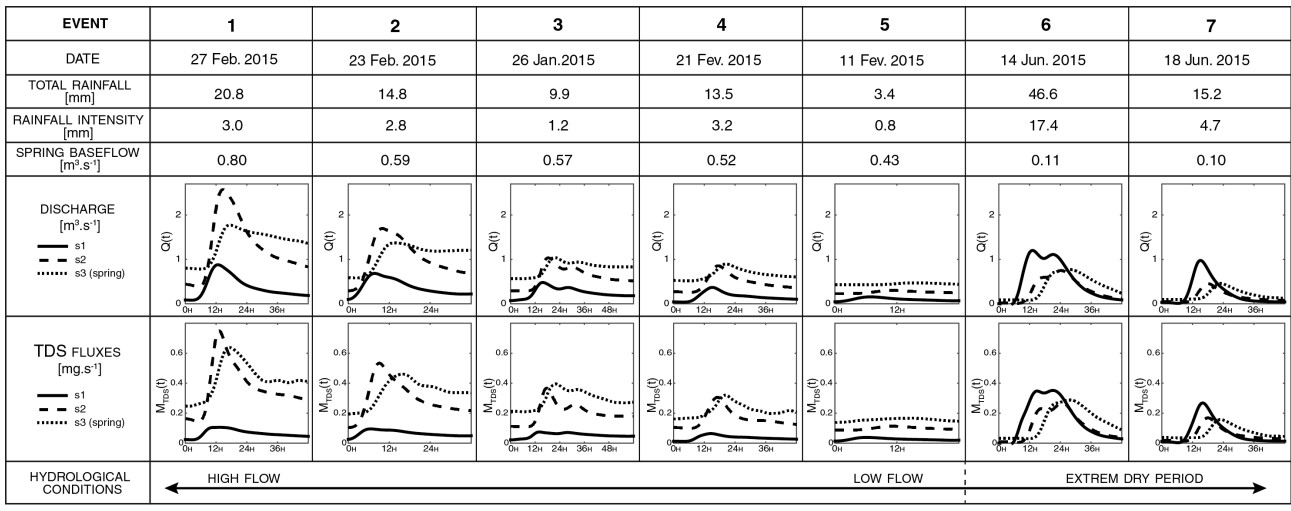

| EVENT | 1 | 2 | 3 | 4 | 5 | 6 | 7 |
|---|---|---|---|---|---|---|---|
| DATE | 27 Feb. 2015 | 23 Feb. 2015 | 26 Jan.2015 | 21 Fev. 2015 | 11 Fev. 2015 | 14 Jun. 2015 | 18 Jun. 2015 |
| TOTAL RAINFALL [mm] | 20.8 | 14.8 | 9.9 | 13.5 | 3.4 | 46.6 | 15.2 |
| RAINFALL INTENSITY [mm] | 3.0 | 2.8 | 1.2 | 3.2 | 0.8 | 17.4 | 4.7 |
| SPRING BASEFLOW [m³.s⁻¹] | 0.80 | 0.59 | 0.57 | 0.52 | 0.43 | 0.11 | 0.10 |

**Figure 4.** Summary of the flood event selection sorted in function of spring baseflow condition.

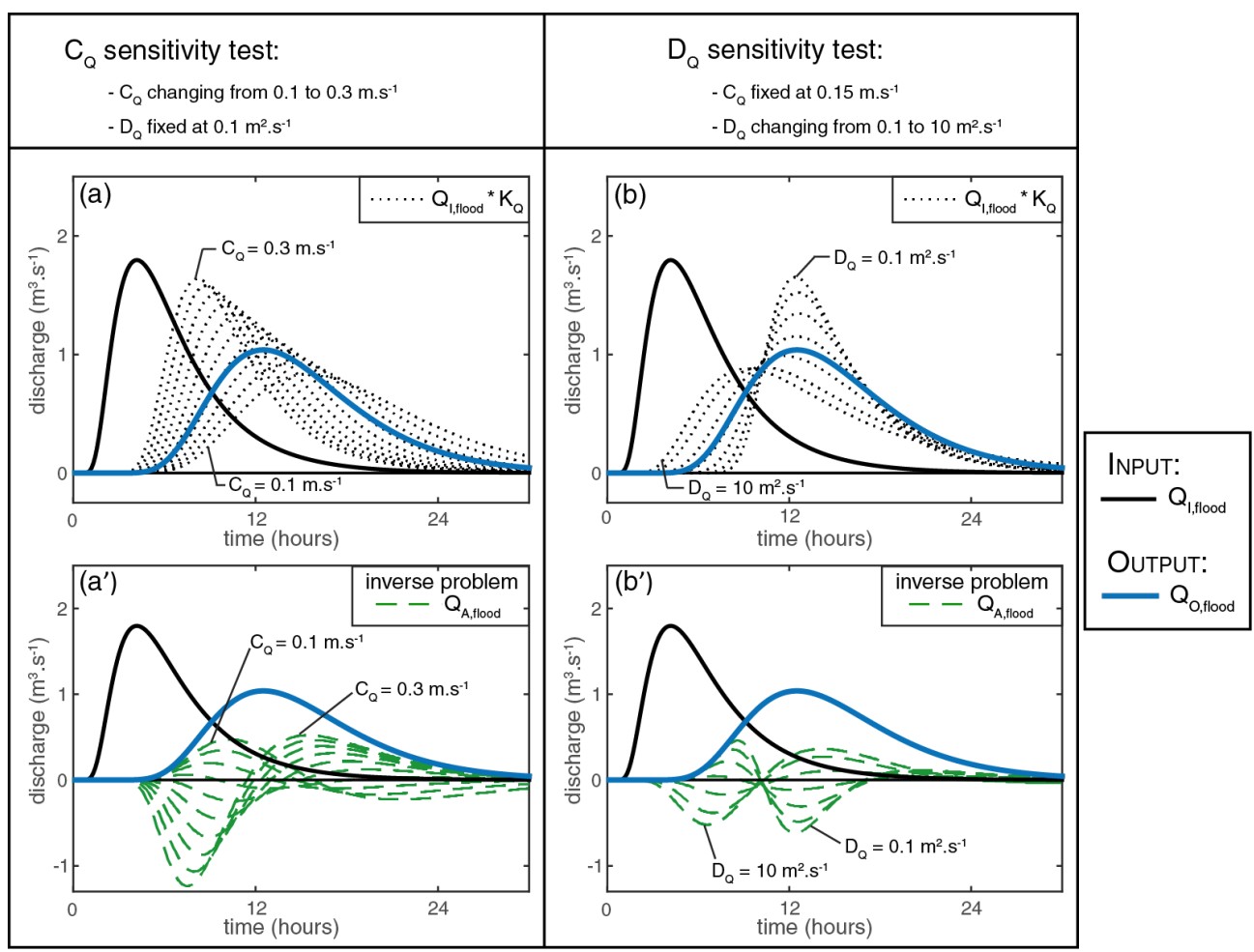

**Figure 5.** Sensitivity analysis of the model parametrization. Graphs (a-b) illustrate the simulation of the routed input $Q_{I,routed}$ (dashed black lines) without lateral exchange, while graphs (a'-b') illustrate the inverse problem approach which simulate lateral flows $Q_{A,flood}$ (dashed green lines). Graphs (a-a') and (b-b') correspond to sensitivity test of $C_Q$ and $D_Q$ , respectively.

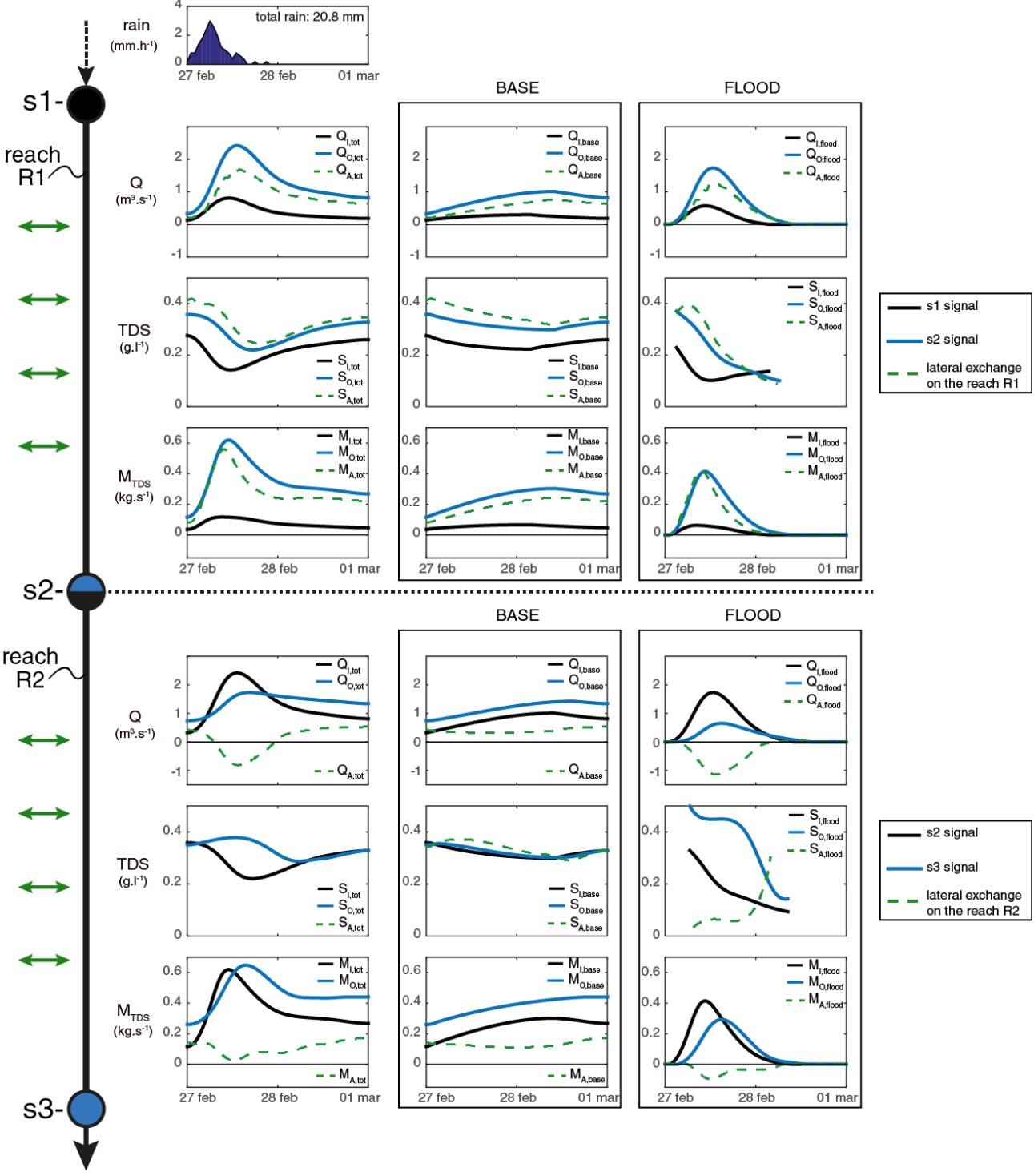

**Figure 6.** Framework application on the event no. 1 in high flow condition along the 2 reaches R1 (top) and R2 (bottom). Total flow and fluxes, base component and flood component, are represented in the first, second, and third column, respectively.

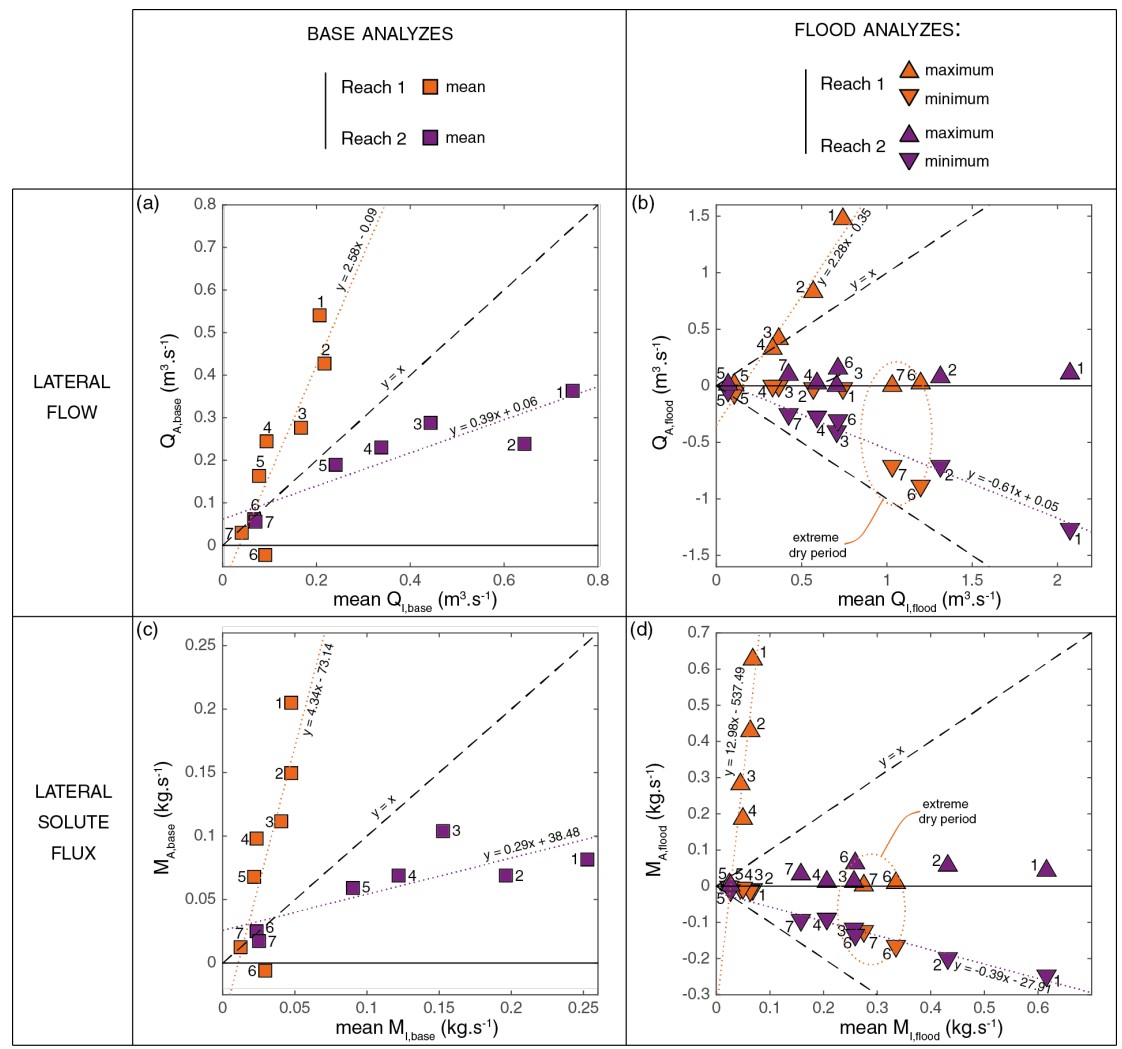

**Figure 7.** Base and flood analyses of the selected event set. Orange and purple symbols correspond to the lateral exchanges modelling along R1 and R2, respectively. Base analyses is performed on mean values ($\square$) exclusively whereas flood analyses take in account minimum ($\triangledown$) and maximum ($\triangle$) values of the calculated lateral exchanges ($Q_{A,flood}$ & $M_{A,flood}$). Base (a) and flood (b) lateral flow ($Q_{A,base}$ & $Q_{A,flood}$) are compared to the mean base and flood flow input ($Q_{I,base}$ & $Q_{I,flood}$) and base (c) and flood (d) lateral solute fluxes ($M_{A,base}$ & $M_{A,flood}$) are compared to the mean base and flood fluxes input ($M_{I,base}$ & $M_{I,flood}$).

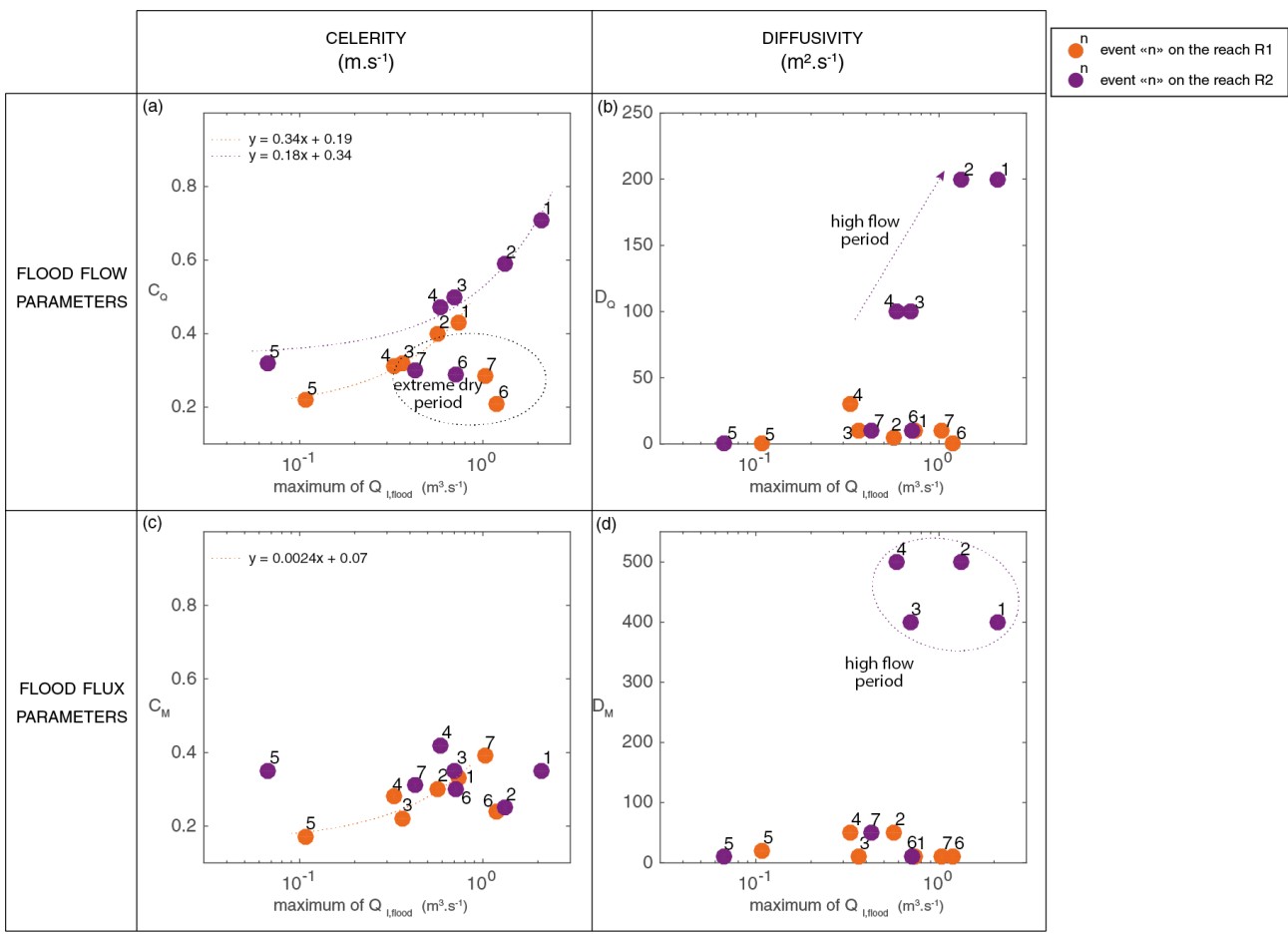

**Figure 8.** Parametrization analyses of the selected event set. Orange and purple symbols correspond to R1 and R2, respectively. Each parameter of the model is compared with the maximum intensity of the flood flow input signal (logarithmic scale).

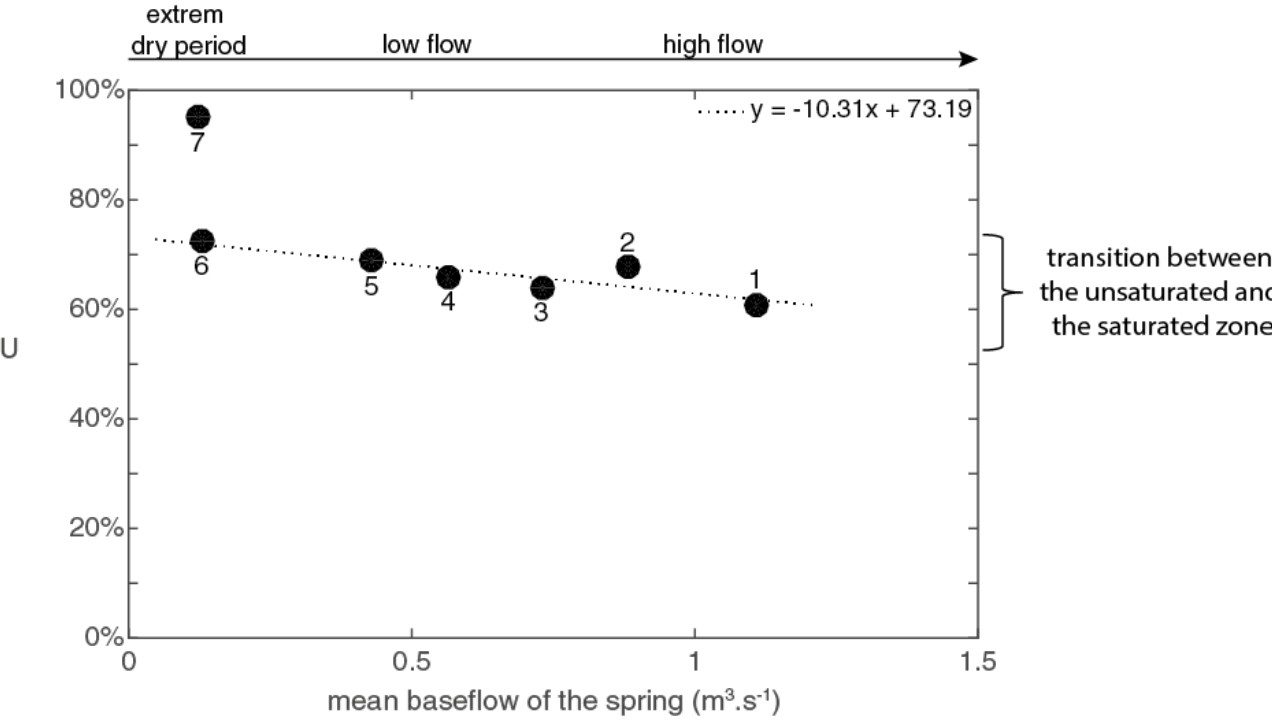

**Figure 9.** Representation of the limit estimation between unsaturated and saturated conduit along R2 in function of the mean spring base flow. U calculation corresponding to the percentage of conduit length of R2 located within the unsaturated zone.

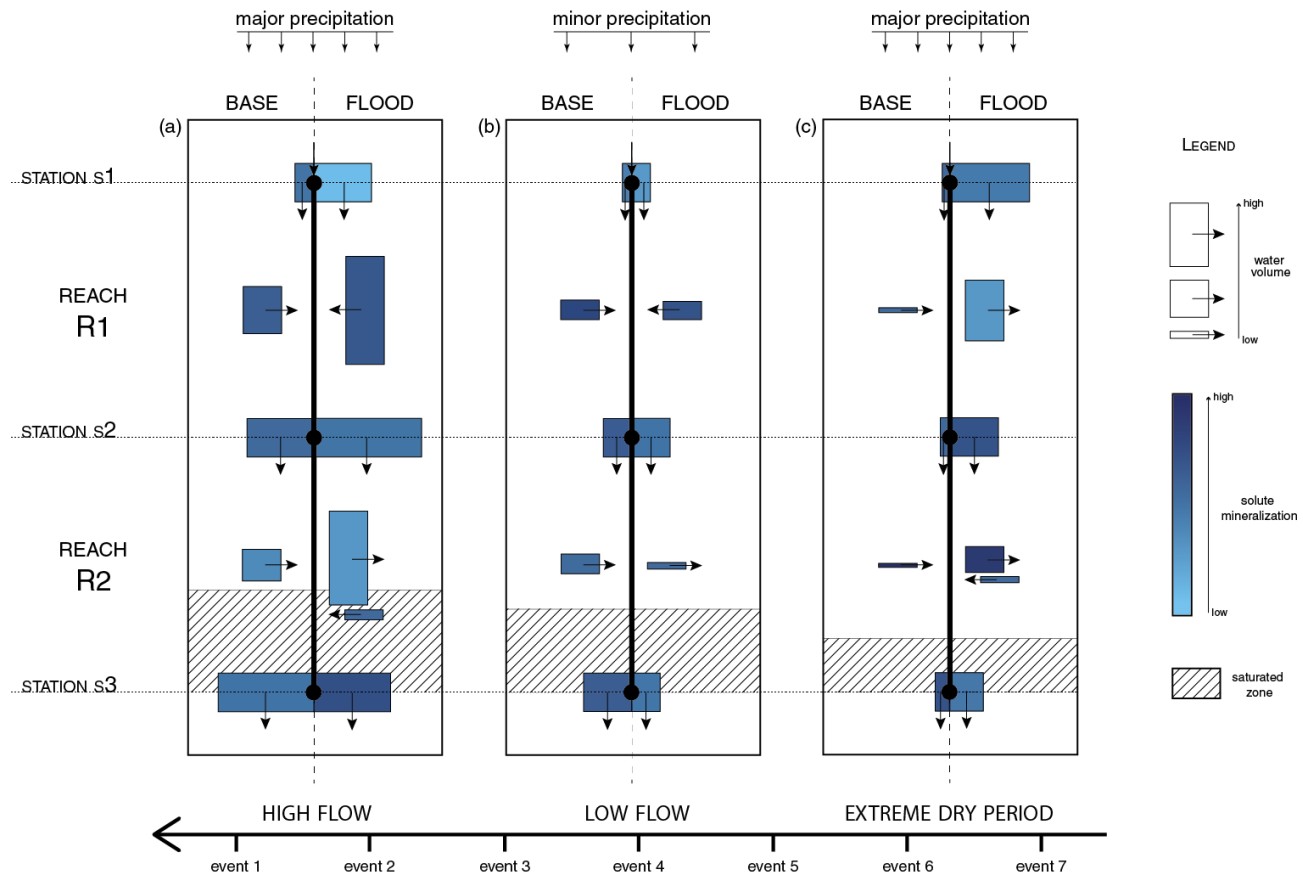

**Figure 10.** Hydrogeological functioning scheme of the Fourbanne karst aquifer, showing the contributions of lateral exchanges in terms of volume and mineralization for 3 hydrological conditions. For each graph, the representation of the lateral exchanges along the reaches (vertical black line) distinguishes the base component (left size) and the flood component (right size).