# Peer review of "Assessing lateral flows and solute transport during floods in a conduit-flow dominated karst system using the inverse problem for advection-diffusion equation"

_Hydrology and Earth System Sciences, 2016_

## Referee Comment (RC1) · Anonymous Referee #1 · 9 Dec 2016

The paper is about the calibration of a diffusive model used to simulate mass and contaminant transport inside a karst conduit network. Authors assume the approximated Saint Venant equations to hold in the reach between two gauged sections, where discharge and concentration are measured, and the aim of the study is the evaluation of the lateral mass and contaminant inflow temporal behavior. Authors apply analytical solutions already provided in the past by Moussa and carry on the calibration by means of a trial-and-errors procedure. The paper has the following limits that, according to my opinion, make questionable the publication of the paper on the Hess journal:

1) The relevance of the analysis is not well explained. If we already know the time series of the input and output mass and concentration fluxes, the average lateral inflow

is simply the difference between input and output average values. Why the temporal variability is so important? 2) The temporal variability of the lateral q flux is strictly related to its spatial variability, which is assumed to be known. In real problem of karst conduits, what is the error of this estimation? In Eq. (6) we find also the spatial derivatives of q. What is the effect of possible spatial discontinuities of the q function? 3) In the application to the study site any validation of the computed lateral flux is missing.

---

## Referee Comment (RC2) · Anonymous Referee #2 · 11 Dec 2016

The referee's comments for the Editor of *Hydrology and Earth System Sciences*
on the paper entitled
FRAMEWORK FOR ASSESSING LATERAL FLOWS DURING FLOODS IN
A CONDUIT-FLOW DOMINATED KARST SYSTEM USING AN INVERSE DIFFUSIVE
MODEL

by Cybèle Cholet, Jean-Baptiste Charlier, Roger Mussa, Marc Steinmann and Sophie Denimal

General comments

In the submitted manuscript the time and space variability of lateral exchanges for flow and dissolved matter in karst conduit network is considered. According to the Authors information "a framework giving new keys ..." is proposed. To my mind the new keys are dealing with application of the advection - diffusion equation for both 1D unsteady flow with free surface and solute transport. However, the Authors assumed a priori linearity of both considered processes. This causes that a superposition is valid (separation of the base flow and the base solute transport can be done) as well as the convolution approach can be applied. In the case of flow such assumption is not obvious as the kinematic wave celerity involved in the diffusive wave equation depends on the unknown discharge. Maybe this requires additional Authors' comment. Consequently, because of the assumed linearity both problems unsteady flow and unsteady solute transport are analyzed using a uniform approach because both problems are described using the same type of equation. When the observations at the upstream end and at the downstream end are known then determination of the lateral inflow/outflow constitutes some kind of inverse problem. The problem is solved using analytical techniques applied to the advection-diffusion transport equation describing both flow and transport.

My general conclusion is as follows: the manuscript is interesting contribution dealing with application of the similar mathematical description in the form of advection – diffusion equations for two different processes for which some kind of inverse problem is solved. However, before the final decision of the Editor, minor revision of the manuscript should be carried out.

Specific comments:

Page 2, line 35 and page 3, line 1:

"... a simplification of the full SVE, and is even a higher order approximation than the uniform formulae (i.e. Manning's formula."

To my mind it is impossible to compare both mentioned cases of flow as they are incomparable. The diffusive wave is dealing with unsteady flow while the Manning formula describes steady uniform flow.

Page 3, lines 7-9

"Combined with Manning's equation or Chézy equation the DW can be simplified to one single equation (Mussa, 1996; ...)".

The diffusive wave equation in the form of advection-diffusion transport equation is derived using original differential continuity equation and the simplified momentum equation only. If we use the continuity equation and the Manning's equation, i.e. the simplified dynamic equation in which only the gravitational and friction forces are taken into account, then one obtains another type of simplified flow equation, namely the kinematic wave equation.

Page 3, lines 10-12

In the sentence presented in these lines is stated that "..., an analytical solution unconditionally stable of the Hayami model exists Mussa (1996)."

This is incorrect because the question of stability or instability of solution of the differential equations is related to the numerical methods applied for their solution but it has nothing in common with the analytical solution.

Page 3, line 20

The Authors use the term "the advection-dispersion equation". It seems to me that it would be better if they used rather the term "advection-diffusion equation" as it is commonly applied in mathematical physics. Note that the term "dispersion" has triple meaning in hydromechanics. One of them is related to the groundwater flow. Regardless on the roots of diffusive term in the transport equation and its physical interpretation, from the mathematical point of view it is the diffusive term.

Page3, lines 24-27

To my mind the presented comment is written imprecisely. Although the diffusive wave equation and the advection-diffusion transport equation are very similar being of the same type, it is worth to remember that they were obtained in completely different ways. The

advection-diffusion transport equation was derived starting from the mass conservation principle applied for matter dissolved in the water and taking into account two basic processes of transport: advection and diffusion in which the Fick's law leading to the diffusive term was applied. As far as the diffusive wave equation is considered, the continuity differential equation and the simplified dynamic equation were combined. The diffusive term appeared as a result of mathematical transformations, not as a flux representing typical physical diffusion. Summarizing, in such a situation it is hard to tell that the diffusive wave equation is applied for solute transport. Both phenomena are treated using the same mathematical approach as the governing equations represent the same type.

It seems to me that the process of "mass propagation" does not exist. Rather the propagating wave causes transport of dissolved matter.

Page 4, lines 14-16

The explanations given in these lines are incorrect. A unique solution of Eq. (1), which is of parabolic type, requires appropriate additional conditions imposed at the limit of the solution domain ($0 \leq x \leq L$ and $t \geq 0$). Solution of Eq. (1) with only one boundary condition, as stated by the Authors, is impossible. Of course, Hayami respected the required conditions. He assumed the following domain of solution: $0 \leq x < \infty$ and $t \geq 0$. The initial condition was: for t=0 Q(x,t)=0 for $x \in \langle 0, \infty)$ whereas, two boundary conditions are as follows: for x=0 $Q(x,t)=\delta(t)$ and for x=X→∞ Q(x, t)=0 is the Dirac delta function. Consequently, he obtained the following solution:

$$Q(x,t) = \frac{1}{2\sqrt{\pi \cdot D}} \frac{x}{t^{3/2}} \exp\left(-\frac{(C \cdot t - x)^2}{4D \cdot t}\right) \qquad (R.1)$$

Since the initial and boundary conditions assumed by Hayami correspond to definition of the impulse response function, then with any open channel reach of length x=L can be related the following impulse response function:

$$K(t) = \frac{1}{2\sqrt{\pi \cdot D}} \frac{L}{t^{3/2}} \exp\left(-\frac{(C \cdot t - L)^2}{4D \cdot t}\right) \qquad (R.2)$$

Note that this equation corresponds to Eq. (5).

On the other hand, each linear dynamic system described by a differential equation can be described alternatively, using the convolution:

$$O(t) = \int_0^t I(t - \tau) \cdot K(\tau) \cdot d\tau \qquad (R.3)$$

where I(t) is the input function, O(t) is the output function whereas $\tau$ is dummy parameter.

Summarizing, the linear dynamic system can be described either by the differential equation or by the convolution. Both representations are equivalent, what means that the downstream hydrograph can be obtained via numerical solution of appropriate differential equation (Eq.1)) or by computation of the convolution integral (Eq. (4) for known kernel function K(t).

My question is following: which reasons decided that instead of direct solution of the diffusive wave equation the Authors preferred using of the convolution approach? It is well known that numerical solution of the linear advection-diffusion equation, particularly when the diffusion is sufficiently strong, is not a problem.

Another question - when the time interval in which the flow is considered is very large, i.e. when time t, being the upper limit of the convolution interval, is increasing while computation then the problem of the system's memory occurs. It is well known, that the memory of real dynamic system is limited and finite so that an input from distant past does not influence the output at the moment. Another speaking, the flow memory corresponds to time elapse of the kernel function. In such a case the convolution (R.3) should be written rather as follows:

$$O(t) = \int_0^p I(t - \tau) \cdot K(\tau) \cdot d\tau \qquad (R.4)$$

where p is a memory of considered system. Certainly the Authors had to face this problem during computations and they had to solve it. It seems to me that a short comment on this question would be interesting for the readers.

Page 5, line 7

It seems to me that Eq. (9) is written incorrectly. If time t is the upper limit of integral then integration of the function q(x, t) cannot be carried out with regard to space x.

Page 6, lines 23-24

If the coefficients $C_Q$ and $D_Q$ corresponds to water flow then the coefficients $C_M$ and $D_M$ should correspond rather to solute transport than to mass fluxes. Similar improper terms are used in other places of the manuscript as well.

Moreover, because $C_Q$ represents the kinematic wave celerity then it can be related to the advection velocity (flow velocity) occurring it the advection – diffusion transport equation. As it is well known, for the Manning formula one has

$$C_Q = \frac{5}{3} C_M .$$

Some results presented in Fig. 8 seem to confirm this relation. This fact allows us reducing of the total number of optimized parameters.

Page 12, lines 29-30

The presented sentence contains the same mistake as discussed above (see Page 3, lines 10-12). The problem of solution stability or instability has nothing in common with the Hayami solution, which is an exact solution of the diffusive wave equation. Moreover, numerical methods introducing numerical instabilities are rather not interesting.

Technical corrections

Page 3, line 12

It seems to me that in this case instead of "Mussa (1996)" it should be rather (Mussa, 1996).

Page3, lines 26

Instead of "by (Mussa, 1996)" it should be rather "by Mussa (1996)".

---

## Referee Comment (RC3) · Anonymous Referee #3 · 20 Dec 2016

Review of "Framework for assessing lateral flows and fluxes during floods in a conduit-flow dominated karst system using an inverse diffusive model" by Cholet et al.

This paper describes the hydrological behaviour of karstic systems that encompass both gravity-driven free-surface flows and pressure-drive conduit flows. A semi-explicit geometry is used to describe the interplay between main reaches (with known curvilinear distances in the streamwise direction) and lateral tributaries (represented by their average "non-point" contributions over selected sections of the main reaches). The chosen methodology is generic enough to explore saturated and non-saturated conditions, base flow and floods, water and suspended particulate matter or any other tracer concerned by the advection-dispersion equation, whose analogy with the diffusive wave model is explored in a quite convincing way, in my opinion. The numerical resolution opted for leans on the analytical resolution of the diffusive wave proposed in former papers by Moussa (1996), Moussa and Bocquillon (1996) and Hayami (1951) which in turn forces the several simplifications in system geometry mentioned in the above. However, it seems to me that the gain in understanding the complexity of the studied karstic systems is worth the relative loss of "lateral" precision, that is, soon as the tributaries are not the key concern, i.e. the main reaches can be properly identified. I am not sure this point has been explicitly addressed but this is certainly a minor concern. I indeed have no major issues with this work and had a good time reading the paper. I therefore recommend it for publication provided the series of questions and remarks listed below receive appropriate answers.

Title

The word "fluxes" is a bit vague, as it is in some places of the manuscript. In this title, "fluxes" could be anything from suspended particulate matter to radioactive or chemical tracers. It sounds like you rather meant mass fluxes here and there in the paper, while strictly speaking any quantity whose movement is described by the advection-dispersion equation may fit in the word "fluxes". Please address this point.

Introduction

P2L1 "rapid transitions" P2L2 "Hauns et al. (2001)" P2L3 "but vanishes at the benefit of an increase in dispersivity with increased distances" P2L4 Does "concentration" mean sediment concentration ? P2L11 "storage exchange fluxes" maybe deserves an explanation for non-specialist readers P2L13 And the same for "gaining and losing reaches" to prevent any misinterpretation P2L17 "the difficulty to model and quantify the spatial patterns of tracer concentrations and..." P2L19 "zones" P2L20 "parameterization" P2L20 I think it is rather "had to account for" than "was strongly impacted by" P2L23 "large-magnitude quick flows" and quick is a bit naive, "flash" may be better here P2L24 "The SVE may be used to assess hydrodynamic processes as they describe..." P2L31

"information" P2L31 "is required" P2L32 degraded mode should write "degraded mode" P2L35 "be considered a valuable simplification" or maybe "relevant" instead of "valuable" P3L1 "SVE while staying a higher order" P3L1 This is interesting and relevant but deserves more indications on the comparative merits and drawbacks of the SVE, the DW, the Kinematic Wave and the uniform formulae. The same question comes a bit later in the paper in situations in which the diffusivity term loses significance and strength before the celerity term. P3L13 By "predetermination" you mean "first guess" or "starting values" to be fitted later? Then this should be made explicit. Even if the convergence to the correct pair of values seems easy to achieve, could we have a few words on the necessity (or not) to provide "good enough" starting values? P3L15 "are part of the hydrological" P3L16 "that was described by Moussa et al." P3L17 "the exchanges with" P3L23 Unwanted line break? P3L24 "Singh (2002)" P3L25 And there this assumption on what "flux" means. If you wish to keep the diversity of meanings possibly covered by "flux" you could make it clear in the abstract, the introduction and the M&M section. P3L26 "Moussa (1996)" P3L28 "coupling water flow"

Modelling approach

P4L7 "is time and the celerity... and diffusivity" P4L11 Please briefly indicate here how the separation is made (I think it is mentioned later) P4L20 What is the origin of t? Does it start at t=0, what is the domain of values for t and what is the associated physical meaning? Think of non-specialists readers here. P4L25 Notation: there should be no "." between units. Are the notations homogeneous throughout the paper? P5L6 "uniform lateral distribution" P5L19 "The classical" P5L21 "lateral fluxes occur" P5L22 "straightforward" for "obvious" P5L24 This phrase on volume conservation seems a bit strange here. Isn't it guaranteed by the assumption that flow is incompressible? P6L1 "mass flux rate" P6L3 Explain what "mass-chemograph" is! P6L11 "By analogy with (5) the kernel function" P6L22 "the required data" P6L23 "includes" P6L24 I suggest another formulation. "Figure 2 gives a graphical representation of this framework whose 7 stages are listed below" P7L1 "from the hydrograph" P7L7 "using two steps." P7L8

"First" P7L11 "results as an automatic optimization with a quick convergence" P7L12 "needs" but why mentioning other cases without additional indications on what theses cases may be? P7L17 "The determination (if possible)... (t) follows then" P7L20 "events" P7L23 "will increase with more pronounced flood peaks"

Study site

P8L17 "15-minute" P9L3 What makes you think the selected event is representative? More generally, how do you define a representative event? The question may seem tricky but the answer depends on what you aim to show with the sensitivity analysis you propose. In fact, I do not think you need to assume there are any representative events as soon as the aim of your sensitivity analysis is to prove your methods are "discriminating" enough, meaning your model is able to identify the key factors in the description of the system. P9L7 I think you should rather state that "CM and DM play similar roles as CQ and DQ" P9L20 "the lower the peak flow intensity" P9L23 "could be reversed"

Results

P10L6 "but that it was due lateral" P10L9 "were half these of" P10L12 "dynamics" P10L13 "were simulated" P10L24 "2.6 times higher" P10L28 You mention a lower slope but is the difference between 0.3 and 0.4 really striking? P11L7 Delete "show"? P11L14 "towards" does not seem appropriate here P11L16 "the distribution of values for the model parameters" P11L17 "intensities, with the aim to retrieve information on flow dynamics" instead of the existing phrasing? P11L18 "contain" rather than "furnish" P11L20 "which span both the unsaturated and the saturated zones" P11L21 You indirectly mention the role of antecedent/initial conditions. I think this point should be better and more explicitly addressed as soon as you intend to provide a rather generic of the hydrological behaviour of the system. In particular, do you think you have enough events to draw general conclusions regarding the sensitivity of the model to its initial conditions. This would be a step towards more genericity and a possible way to endow

the model with increased predictive capabilities. P11L24 "did not find" P11L28 "again with high values only for events" P12L2 "trends" rather than "patterns" P12L3 "for the hydrological" P12L8 "therefore deliver" P12L11 Does this hypothesize no effect exist on DQ and/or does this assumes DQ is definitely a second-order term in such cases? If so, this could be voiced here. P12L16 Where does this approximation come from?

Discussion and conclusion

This section should be separated into "discussion" and "conclusion". Moreover, I tend to disagree with the rather atypical names of the sections in this paper. According to me, "Study site" should be the 2.2 part of a Section 2 named "Material and methods", following the 2.1 part entitled "Modelling approach".

I found the discussion repetitive in some parts, which I suggest to remove. This would facilitate the reader's task and make the additional indications (i.e. the real discussion elements) more punchy.

Delete most of P12L20 to P13L2 P13L4 "variability" is repeated The discussion P13L7-16 is very instructive P13L18-19 What would be the effect of a smaller spatial discretization in the streamwise direction? Could the model handle a smaller dx and could this be part of the discussion? P13L29 Alternatively what are the requirements for boundary conditions not to be poor? What are the required BC to ensure the model performs well (or to make sure the BC are not the weak point of the description)? Delete P13L31-34 and start the next phrase with "The functional scheme..." P14L3 "Fig" not "fig" and in other occurrences too P14L6 "had lateral inflows" P14L7 "these mineralized inputs" P14L11 "mineralized" P14L19 Delete "derived" P14L31 "R1" P15L1-7 Make it shorter or delete P15L8+ The last part of the document is convincing and well-written. Should be the conclusion and the message for future research leads.

---

## Referee Comment (RC4) · Anonymous Referee #4 · 27 Dec 2016

The manuscript introduces a new framework which enables assessment of inflows/outflows to/from the channel reach in a karst aquifer. The paper presents a nice combination of field and modelling work and is as such worth publishing. My comments are mostly related to the presentation of concepts (and to some extent results) which should be made clearer.

Here is the list of comments (P#L#) refers to the page and line number to which the comment refers to the page and line number the comment refers to:

The concept of Diffusive wave equation was first introduced at P2L35 and reintroduced at P3L5. Restructure these paragraphs.

P3L8 A statement that DW is used for mass transport is a bit misleading. Although the equation is the same as the ADE, it is built on different conservation principles and driving forces. The statement might confuse a reader. The same comment goes to P327.

P4L15: What do you mean with **without downstream boundary conditions**. Please clear up.

P4L 14-17: This paragraph is somehow awkward. What do you mean with **direct model** ? The $Q_{i,routed}$ is introduced, but not told what it represents; this would be helpful for someone not familiar with the older literature... The aim of 2.1.1. is somehow lost until P4L21: I miss an earlier clear statement that $C_Q$ and $D_Q$ are looked for. How are $Q_{I,routed}$ and $Q_{O,flood}$ compared?

P5L8 (and before): It looks like that the model allows assessment of *q(x,t)*, however as far as I understanding it allows only time dependence of the total in/out-flow between I and O. In other words, spatial distribution of in/out-flows is not obtained?

P5EQ10 and EQ11: Change $A_Q(t)$ to $Q_A(t)$.

P5L20-25: The paragraph is hard to read. 1) What do you mend by Âżapplication of ADE is not so obviousÂń. Does this mean that equation itself is Âżnot obviousÂń or that the solution is not straightforward. ... If I understand correctly, the idea is to determine lateral flow from DW (previous chapter) and introduce it into ADE to finally get $M(t)$ and $S(t)$. Why introducing $Q(t)$ in Equation 15 ? Is this $Q_{A(t)}$ ? I miss clear presentation of ideas and concepts. It would be good if you put in the ADE equation.

P9L12-20: Can one infer from DW equation that sensitivity to celerity is orders of magnitude higher than sensitivity to diffusivity?

---

## Author Comment (AC1) · 10 Jan 2017

**"Framework for assessing lateral flows and fluxes during floods in a conduit-flow dominated karst system using an inverse diffusive model"**

by C. Cholet, J.-B. Charlier, R. Moussa, M. Steinmann, and S. Denimal

(Hydrol. Earth Syst. Sci. Discuss., doi:10.5194/hess-2016-565, 2016)

We are very grateful to the four reviewers for their constructive comments of the manuscript. We totally agree with all their recommendations. Please find below our responses (in blue) to the comments (in black) of all four reviewers.

**Author comment to Referee #1**

*The paper is about the calibration of a diffusive model used to simulate mass and contaminant transport inside a karst conduit network. Authors assume the approximated Saint Venant equations to hold in the reach between two gauged sections, where discharge and concentration are measured, and the aim of the study is the evaluation of the lateral mass and contaminant inflow temporal behavior. Authors apply analytical solutions already provided in the past by Moussa and carry on the calibration by means of a trial-and-errors procedure.*

*The paper has the following limits that, according to my opinion, make questionable the publication of the paper on the Hess journal:*

*R1 - Comment 1) The relevance of the analysis is not well explained. If we already know the time series of the input and output mass and concentration fluxes, the average lateral inflow is simply the difference between input and output average values. Why the temporal variability is so important?*

Maybe the text was not sufficiently explicit and clear. Most karst systems are only accessible punctually, leaving large portions of the system inaccessible for direct observation. Our modelling framework is a tool to decipher the hydrological functioning of such inaccessible conduits located between 2 monitoring stations. As mentioned by the referee, water and mass balance of lateral exchanges can be estimated simply by the difference between input and output average values. Thus, the interest of simulating temporal variability is to characterize the evolution of lateral flows and their mineralisation during the flood. This is important because we showed for instance that these exchanges can be successively positive or negative during a single flood. In other words, we present evidence for the existence of a complex dynamic of lateral exchanges occurred, knowing that such processes cannot be identified without a temporal analysis. Figures 6 and 7 illustrate this point by presenting contrasted evolutions of discharge and fluxes showing the additional description that can be done during a flood event in comparison to average values.

Another interest to simulate the temporal variability is discussed in Section 5.2 where we showed that the approach may also help to characterize interaction with the surrounding rock matrix. For instance, we are able to model lateral exchanges occurring in the saturated zone that can be related to reversal flow, which is coherent with concepts of conduit/matrix interactions, mentioned by several authors in the literature.

Following the referee comment, in order to better clarify the relevance of the analysis of lateral exchange temporal variability, we will add complementary explanations mainly in the "Introduction" and in the section "2.3 Framework" to explain how the model simulations

describing the temporal variability of the lateral exchanges are used to better characterize interactions along the conduit during flood and the hydrogeological functioning of the karst aquifer.

*R1 - Comment 2) The temporal variability of the lateral q flux is strictly related to its spatial variability, which is assumed to be known. In real problem of karst conduits, what is the error of this estimation?*

We agree with the interrogation of the reviewer concerning the impact of the hypothesis used on the lateral flux spatial distribution on the error of the estimation of these fluxes.

In fact, the Hayami analytical solution of the diffusive wave model assumes a uniformly distributed flow rate $q(x, t)$ along the reach (see Section 2.1.2 for the mathematical description). The hypothesis of a uniformly spatial distribution along the reach is the simplest hypothesis when non additional information from the field is available. Moreover, under this hypothesis, and under the hypotheses used in the Hayami model, an analytical solution of the inverse model easy-to-use exists. Thus, the unknown $q(x, t)$ is reduced to $q(t)$. Even if the spatial distribution of $q(x, t)$ is unknown, the simulated $q(t)$ under the hypothesis of uniformly spatial distribution will give to the modeller important information on the temporal variability of lateral exchanges, because it enables to distinguish three cases: i) negative $q(t)$ during the whole event; ii) positive $q(t)$ during the whole event; iii) alternating positive and negative $q(t)$ during an event. It enables also to calculate the temporal distribution of lateral flow $q(t)$, and the maximum/minimum values of $q(t)$. Moreover, as the model is applied for two different variables, the flow and the solute transport, two different $q(t)$ for the flow and the solute will be available for the modeller, which can be crossed with other field information in order to identify a functioning scheme of the karst system.

Moreover, if additional information on lateral fluxes is available, as for example punctual inputs/outputs on the reach, information on the amplitude of the spatial distribution of lateral flows, or measurements of solute concentrations on different points, the framework proposed in the paper is generic and can be easily used. In this case, the studied zone has to be subdivided into different reaches with eventually punctual inputs/outputs on some nodes; then the inverse model can be applied on each reach. An estimation of the error induced can be obtained by applying the model on a reach under different hypotheses for spatial distribution of lateral flow.

The study case in the paper corresponds to the case generally encountered in practice with two reaches where the spatial variability is unknown. However the geometry of the main conduit is globally well-known in the unsaturated zone (see the map on Figure 3), but the precise localization of tributaries and losses is unknown in detail. Moreover, their contributions over time are totally unknown. In this context, our approach is precisely used to make a diagnostic of these lateral exchanges, which are generally impossible to know in such heterogeneous media.

We will add additional explanation concerning the hypotheses of the model and the error induced in the Section 2.3 "Framework".

*In Eq. (6) we find also the spatial derivatives of q. What is the effect of possible spatial discontinuities of the q function?*

Yes, Eq. (6) gives the general form of the diffusive wave equation for any spatio-temporal distribution of $q(x, t)$. However Eqs. (7 and 8) give the resolution of Eq. (6) in the particular case of uniformly spatial distribution of $q(x, t)$ along the reach, under the hypotheses used in the Hayami model (C and D constant). We will add an explanation in the text.

*R1 – Comment 3) In the application to the study site any validation of the computed lateral flux is missing.*

The objective of the paper is to present a framework in the general case where only flow (and eventually solute concentration) is measured on gauging stations without any additional information on lateral flow. The aim is to use simultaneously both measured input and output hydrographs (and eventually input/output solute concentrations) in order to identify lateral inflow/outflow temporal distribution. Contrary to classical modelling approach where the measured output hydrograph is used only to validate a model, the inverse modelling approach developed herein uses all information available in both input/output data. Hence, our modelling approach is not used with data sets allowing a validation of the computed lateral fluxes. In the study case, there is no monitoring of the tributaries/losses along the 2 reaches of the study site. But due to the complex geometry of the conduit network, this is something probably impossible in a large karst system. And this is because such data are almost impossible to get that we propose to apply some such modelling approaches to investigate lateral contributions.

However, if additional variables are measured, as for example piezometer levels or hydrographs on tributaries (or concentrations in the water table or tributaries), a validation can be undertaken by comparing the measured variable to the simulated lateral flow hydrograph. It has been proposed by Charlier et al. (2015), in which the hydrograph dynamics of lateral springs is compared with the simulated lateral exchanges.

An additional description in the manuscript will be added in the Section "2.3 Framework" in order to discuss the added-value when using the inverse model for identifying lateral flow without any validation. We will discuss also the possibility to use additional variables to validate the model and propose complementary measurements.

**Author comment to Referee #2**

*General comments*

*In the submitted manuscript the time and space variability of lateral exchanges for flow and dissolved matter in karst conduit network is considered. According to the Authors information "a framework giving new keys ..." is proposed. To my mind the new keys are dealing with application of the advection - diffusion equation for both 1D unsteady flow with free surface and solute transport. However, the Authors assumed a priori linearity of both considered processes. This causes that a superposition is valid (separation of the base flow and the base solute transport can be done) as well as the convolution approach can be applied. In the case of flow such assumption is not obvious as the kinematic wave celerity involved in the diffusive wave equation depends on the unknown discharge. Maybe this requires additional Authors' comment.*

*Consequently, because of the assumed linearity both problems unsteady flow and unsteady solute transport are analyzed using a uniform approach because both problems are described using the same type of equation. When the observations at the upstream end and at the downstream end are known then determination of the lateral inflow/outflow constitutes some kind of inverse problem. The problem is solved using analytical techniques applied to the advection-diffusion transport equation describing both flow and transport.*

*My general conclusion is as follows: the manuscript is interesting contribution dealing with application of the similar mathematical description in the form of advection – diffusion equations for two different processes for which some kind of inverse problem is solved. However, before the final decision of the Editor, minor revision of the manuscript should be carried out.*

Thank you!

We agree that the new keys are dealing with application of the advection - diffusion equation for both 1D unsteady flow with free surface and solute transport. First, this hypothesis makes the separation/superposition of the base flow/surface runoff for both flow and solute transport valid as well as the application of the convolution approach. Second, the hypothesis of linearity of both considered processes enables to have an analytical easy-to-use solution of the inverse model. This is one major advantage of the Hayami resolution of the diffusive wave model which is not possible when using the kinematic wave model where celerity depends on unknown discharge. Additional comments will be added in Section 2 "Modelling approach".

*Specific comments:*

*R2 – Comment 1) Page 2, line 35 and page 3, line 1:*

*"... a simplification of the full SVE, and is even a higher order approximation than the uniform formulae (i.e. Manning's formula."*

*To my mind it is impossible to compare both mentioned cases of flow as they are incomparable. The diffusive wave is dealing with unsteady flow while the Manning formula describes steady uniform flow.*

We agree with the comment. The sentence will be corrected: "… *the diffusive wave equation can be considered as a simplification of the full SVE. This approach* …"

*R2 – Comment 2) Page 3, lines 7-9*

*"Combined with Manning's equation or Chézy equation the DW can be simplified to one single equation (Mussa, 1996; ...)".*

*The diffusive wave equation in the form of advection-diffusion transport equation is derived using original differential continuity equation and the simplified momentum equation only. If we use the continuity equation and the Manning's equation, i.e. the simplified dynamic equation in which only the gravitational and friction forces are taken into account, then one obtains another type of simplified flow equation, namely the kinematic wave equation.*

We agree. The sentence will be modified to explain how the diffusive wave equation (and the advection-diffusion transport equation) is derived combining the differential continuity equation and the simplified momentum equation, when the acceleration terms in the momentum equation can be neglected.

*R2 – Comment 3) Page 3, lines 10-12*

*In the sentence presented in these lines is stated that "..., an analytical solution unconditionally stable of the Hayami model exists Mussa (1996)."*

*This is incorrect because the question of stability or instability of solution of the differential equations is related to the numerical methods applied for their solution but it has nothing in common with the analytical solution.*

We agree. The sentence will be clarified explaining that, in comparison to numerical methods, the advantage of the Hayami solution is an easy-to-use analytical solution for both the flood routing model and the inverse model.

*R2 – Comment 4) Page 3, line 20*

*The Authors use the term "the advection-dispersion equation". It seems to me that it would be better if they used rather the term "advection-diffusion equation" as it is commonly applied in mathematical physics. Note that the term "dispersion" has triple meaning in hydromechanics. One of them is related to the groundwater flow. Regardless on the roots of diffusive term in the transport equation and its physical interpretation, from the mathematical point of view it is the diffusive term.*

Ok. The term "*advection-dispersion equation*" is largely used in groundwater flow and karst hydrology literature. But we agree with the comment and will replace the term "*dispersion*" by "*diffusion*".

*R2 – Comment 5) Page3, lines 24-27*

*To my mind the presented comment is written imprecisely. Although the diffusive wave equation and the advection-diffusion transport equation are very similar being of the same type, it is worth to remember that they were obtained in completely different ways. The advection-diffusion transport equation was derived starting from the mass conservation principle applied for matter dissolved in the water and taking into account two basic processes of transport: advection and diffusion in which the Fick's law leading to the diffusive term was applied. As far as the diffusive wave equation is considered, the continuity differential equation and the simplified dynamic equation were combined. The diffusive term*

*appeared as a result of mathematical transformations, not as a flux representing typical physical diffusion. Summarizing, in such a situation it is hard to tell that the diffusive wave equation is applied for solute transport. Both phenomena are treated using the same mathematical approach as the governing equations represent the same type. It seems to me that the process of "mass propagation" does not exist. Rather the propagating wave causes transport of dissolved matter.*

We understand the commentary that some terminology used in the manuscript to describe our modelling approach is sometimes confusing. In fact, the diffusive wave equation and the advection-diffusion transport equation have very similar mathematical equation, but of course do not describe the same processes. Additional details will be added to show the differences in the physical basis of each of the diffusive wave equation and the advection-diffusion transport equation as proposed by the Reviewer: i) the Saint-Venant equations (continuity and momentum) for the diffusive wave model; ii) the advection-diffusion equation derived from the mass conservation principle applied for matter dissolved in the water and taking into account two basic processes of transport: advection and diffusion in which the Fick's law leading to the diffusive term was applied. Under some hypotheses, the physical equations of both models can lead to the same similar mathematical equation which can justify the use of the same resolution approach with the Hayami model. We agree that the term "mass propagation" is not appropriate and will replace it by the physical terms, either "diffusive wave equation" or "advection-diffusion equation". We will discuss also the domain and limits of application of the hypotheses and approximations used.

*R2 – Comment 6) Page 4, lines 14-16*

*The explanations given in these lines are incorrect. A unique solution of Eq. (1), which is of parabolic type, requires appropriate additional conditions imposed at the limit of the solution domain ($0 \leq x \leq L$ and $t \geq 0$). Solution of Eq. (1) with only one boundary condition, as stated by the Authors, is impossible. Of course, Hayami respected the required conditions. He assumed the following domain of solution: $0 \leq x < \infty$ and $t \geq 0$. The initial condition was: for t=0 Q(x,t)=0 for x ϵ ⟨0, ∞) whereas, two boundary conditions are as follows: for x=0 Q(x,t)=δ(t) and for x=X→∞ Q(x, t)=0 is the Dirac delta function. Consequently, he obtained the following solution:*

$$Q(x,t) = \frac{1}{2\sqrt{\pi.D}}\frac{x}{t^{3/2}}exp\left(-\frac{(C.t-x)^2}{4D.t}\right) \tag{R.1}$$

*Since the initial and boundary conditions assumed by Hayami correspond to definition of the impulse response function, then with any open channel reach of length x=L can be related the following impulse response function:*

$$K(t) = \frac{1}{2\sqrt{\pi.D}}\frac{L}{t^{3/2}}exp\left(-\frac{(C.t-x)^2}{4D.t}\right) \tag{R.2}$$

*Note that this equation corresponds to Eq. (5).*

*On the other hand, each linear dynamic system described by a differential equation can be described alternatively, using the convolution:*

$$O(t) = \int_0^t I(t-T).K(T).dT \tag{R.3}$$

*where I(t) is the input function, O(t) is the output function whereas T is dummy parameter. Summarizing, the linear dynamic system can be described either by the differential equation or by the convolution. Both representations are equivalent, what means that the downstream*

*hydrograph can be obtained via numerical solution of appropriate differential equation (Eq.1)) or by computation of the convolution integral (Eq. (4) for known kernel function K(t).*

*My question is following: which reasons decided that instead of direct solution of the diffusive wave equation the Authors preferred using of the convolution approach? It is well known that numerical solution of the linear advection-diffusion equation, particularly when the diffusion is sufficiently strong, is not a problem.*

Thank you for the detailed explanation. We totally agree, and will add adequate explanation in the text concerning the initial condition (t= 0; Q(x,t) = 0) and the two boundary conditions: i) upstream for x=0 Q(x,t)=δ(t) (Dirac function) and for x=X→∞ Q(x, t)=0. We will discuss the advantages/disadvantages when using either a numerical solution of the equation or the use of a convolution.

We decide to use a convolution instead of a numerical solution of the linear advection-diffusion equation for two reasons: i) an analytical solution of the inverse model is available and easy-to-use; hence it is more convenient to use the same form of the resolution for both the routing model and the inverse model; ii) the use of a convolution enables to avoid the need for choosing adequate space and time steps in numerical methods (subdividing the reach into space steps dx, and the time into time steps dt) which may introduce numerical instabilities.

*R2 – Comment 7) Another question - when the time interval in which the flow is considered is very large, i.e. when time t, being the upper limit of the convolution interval, is increasing while computation then the problem of the system's memory occurs. It is well known, that the memory of real dynamic system is limited and finite so that an input from distant past does not influence the output at the moment. Another speaking, the flow memory corresponds to time elapse of the kernel function. In such a case the convolution (R.3) should be written rather as follows:*

$$O(t) = \int_0^p I(t - T).K(T).dT \hspace{4cm} (R.4)$$

*where p is a memory of considered system. Certainly the Authors had to face this problem during computations and they had to solve it. It seems to me that a short comment on this question would be interesting for the readers.*

We totally agree with this comment. As there is no problem of time of calculation, the term "p" corresponding to the memory of the system was chosen so large in comparison to the travel time on a channel reach. Moreover, the time step dt in the integral in Eq (R.4) was chosen very small. We will add a comment in the Section 2 "Modelling approach" and give numerical values in the applications.

*R2 – Comment 8) Page 5, line 7*

*It seems to me that Eq. (9) is written incorrectly. If time t is the upper limit of integral then integration of the function q(x, t) cannot be carried out with regard to space x.*

We totally agree with the comment. The upper limit of Eq.(9) is not time, but the length of the reach L. It's an error from our part. The correct equation is:

$$Q_{A,flood}(t) = \int_0^L q(x,t)dx$$

*R2 – Comment 9) Page 6, lines 23-24*

*If the coefficients $C_Q$ and $D_Q$ corresponds to water flow then the coefficients $C_M$ and $D_M$ should correspond rather to solute transport than to mass fluxes. Similar improper terms are used in other places of the manuscript as well.*

*Moreover, because $C_Q$ represents the kinematic wave celerity then it can be related to the advection velocity (flow velocity) occurring it the advection – diffusion transport equation. As it is well known, for the Manning formula one has*

$$C_Q = \frac{5}{3} C_M$$

*Some results presented in Fig. 8 seem to confirm this relation. This fact allows us reducing of the total number of optimized parameters.*

We agree that the term « mass flux » is not appropriate. We will homogenize all notations in the paper such as $C_Q$ and $D_Q$ correspond to water flow, and $C_M$ and $D_M$ correspond rather to solute transport.

Thank you for the comments concerning an eventual relationship between $C_Q$ and $C_M$. Additional discussion will be added concerning the approximation for rectangular sections when using the Manning Equation $C_Q = (5/3) C_M$ or more precisely (Chow, 1959; Moussa and Bocquillon, 1995)

$$C_Q = \frac{dQ}{dA} = \left[\frac{5}{3} - \frac{4}{3} \frac{y}{B + 2y}\right] C_M$$

Where A is the cross sectional area of the flow, B is the cross-sectional width of a rectangular section and y the flow depth. Effectively, Fig. 8 shows that the ratio $C_Q/C_M$ ranges between 1.1 and 2.3. Fixing $C_Q = (5/3) C_M$ enables to reduce the number of parameters to be calibrated from 4 ($C_Q$, $D_Q$, $C_M$ and $D_M$) to 3. Additional discussion will be added in the revised version.

*R2 – Comment 10) Page 12, lines 29-30*

*The presented sentence contains the same mistake as discussed above (see Page 3, lines 10-12). The problem of solution stability or instability has nothing in common with the Hayami solution, which is an exact solution of the diffusive wave equation. Moreover, numerical methods introducing numerical instabilities are rather not interesting.*

Ok. We will make the correction.

*Technical corrections*

*Page 3, line 12*

*It seems to me that in this case instead of "Mussa (1996)" it should be rather (Mussa, 1996).*

Ok. The reference will be corrected (please note that the name is "Moussa" and not "Mussa").

*Page3, lines 26*

*Instead of "by (Mussa, 1996)" it should be rather " by Mussa (1996)".*

Ok. The reference will be corrected.

**Author comment to Referee #3**

*This paper describes the hydrological behaviour of karstic systems that encompass both gravity-driven free-surface flows and pressure-drive conduit flows. A semi-explicit geometry is used to describe the interplay between main reaches (with known curvilinear distances in the streamwise direction) and lateral tributaries (represented by their average "non-point" contributions over selected sections of the main reaches). The chosen methodology is generic enough to explore saturated and non-saturated conditions, base flow and floods, water and suspended particulate matter or any other tracer concerned by the advection-dispersion equation, whose analogy with the diffusive wave model is explored in a quite convincing way, in my opinion. The numerical resolution opted for leans on the analytical resolution of the diffusive wave proposed in former papers by Moussa (1996), Moussa and Bocquillon (1996) and Hayami (1951) which in turn forces the several simplifications in system geometry mentioned in the above. However, it seems to me that the gain in understanding the complexity of the studied karstic systems is worth the relative loss of "lateral" precision, that is, soon as the tributaries are not the key concern, i.e. the main reaches can be properly identified. I am not sure this point has been explicitly addressed but this is certainly a minor concern. I indeed have no major issues with this work and had a good time reading the paper. I therefore recommend it for publication provided the series of questions and remarks listed below receive appropriate answers.*

Thank you!

We totally agree that the gain in understanding the complexity of the studied karst system was not sufficiently addressed as also mentioned by Referee #1. We will address this point in the revised version as given in our response to Comments 1 and 2 of the Referee #1.

**Title**

*R3 – Comment 1) The word "fluxes" is a bit vague, as it is in some places of the manuscript. In this title, "fluxes" could be anything from suspended particulate matter to radioactive or chemical tracers. It sounds like you rather meant mass fluxes here and there in the paper, while strictly speaking any quantity whose movement is described by the advection-dispersion equation may fit in the word "fluxes". Please address this point.*

We agree with the comment and we propose to change in the title the word "fluxes" by "solute transport" to describe the paper more precisely. We will also attach more attention about the use of this term all along the paper.

The title will be modified as follow: "*Framework for assessing lateral flows and solute transport during floods in a conduit-flow dominated karst system using an inverse diffusive model*"

**Introduction**

*R3 – Comment 2) P2L1 "rapid transitions" / P2L2 "Hauns et al. (2001)" / P2L3 "but vanishes at the benefit of an increase in dispersivity with increased distances" / P2L11 "storage exchange fluxes" maybe deserves an explanation for non-specialist readers / P2L13 And the same for "gaining and losing reaches" to prevent any misinterpretation. / P2L17 "the difficulty to model and quantify the spatial patterns of tracer concentrations and …" / P2L19 "zones" / P2L20 "parameterization" / P2L20 I think it is rather "had to account for" than "was strongly impacted by" / P2L23 "large-magnitude quick flows" and quick is a bit naive, "flash" may be*

*better here / P2L24 "The SVE may be used to assess hydrodynamic processes as they describe..." / P2L31 "information" / P2L31 "is required" / P2L35 "be considered a valuable simplification" or maybe "relevant" instead of "valuable" / P3L1 "SVE while staying a higher order" / P3L15 "are part of the hydrological" / P3L16 "that was described by Moussa et al." / P3L17 "the exchanges with" / P3L23 Unwanted line break? / P3L24 "Singh (2002)" / P3L26 "Moussa (1996)" / P3L28 "coupling water flow"*

Thank you for these corrections. We will take into account all these remarks in the revised version.

*R3 – Comment 3) P2L4 Does "concentration" mean sediment concentration?*

It doesn't correspond to sediment concentration, but solute concentrations. It will be precise in the new version of the manuscript.

*R3 – Comment 4) P2L32 degraded mode should write "degraded mode"*

It seems there is a missing word in the comment. We actually used the terminology described in Le Moine et al. (2008).

*R3 – Comment 5) P3L1 This is interesting and relevant but deserves more indications on the comparative merits and drawbacks of the SVE, the DW, the Kinematic Wave and the uniform formulae.*

In agreement with the Comment 1 of the Reviewer #2, the sentence will be corrected in the revised version of the manuscript as follows: "… *the diffusive wave equation can be considered as a simplification of the full SVE. This approach* …". In fact, the diffusive wave model and the uniform formulae cannot be compared because they describe two distinct flow processes (unsteady flow and steady uniform flow, respectively).

*R3 – Comment 6) The same question comes a bit later in the paper in situations in which the diffusivity term loses significance and strength before the celerity term.*

The comparison between diffusivity and celerity terms is also discussed in the sensitivity analyses. However, in the aim to define the diffusive term more precisely, additional description will be added at the end of the introduction.

*R3 – Comment 7) P3L13 By "predetermination" you mean "first guess" or "starting values" to be fitted later? Then this should be made explicit. Even if the convergence to the correct pair of values seems easy to achieve, could we have a few words on the necessity (or not) to provide "good enough" starting values?*

The term "predetermination" means that before the application of the inverse model, the parameters are already optimized by a trial-and-error method using the direct modelling approach. Whatever the starting values used, the optimization will provide progressively suitable pair of values for the application of the inverse model. We will modify the sentence to make it more clear in the new version of the manuscript. P3L13: "*The inverse model needs as input the inflow hydrograph, the outflow hydrograph together with the celerity and diffusivity values already optimized in the direct model*".

*R3 – Comment 8) P3L25 And there this assumption on what "flux" means. If you wish to keep the diversity of meanings possibly covered by "flux" you could make it clear in the abstract, the introduction and the M&M section.*

We totally agree with this comment also given by Reviewer #2 (Comment 5). Our terminology about flux maybe be awkward in some part of the paper; the new formulation about "solute mass flux" will be homogenized in the revised version of the manuscript.

Modelling approach

*R3 – Comment 9) P4L7 "is time and the celerity... and diffusivity" / P4L25 Notation: there should be no "." between units. Are the notations homogeneous throughout the paper? / P5L6 "uniform lateral distribution" / P5L19 "The classical" / P5L21 "lateral fluxes occur" / P5L22 "straightforward" for "obvious" / P6L1 "mass flux rate" / P6L3 Explain what "mass-chemograph" is! / P6L11 "By analogy with (5) the kernel function" / P6L22 "the required data" / P6L23 "includes" / P6L24 I suggest another formulation. "Figure 2 gives a graphical representation of this framework whose 7 stages are listed below" / P7L1 "from the hydrograph" / P7L7 "using two steps." / P7L8 "First" / P7L11 "results as an automatic optimization with a quick convergence" / P7L12 "needs" but why mentioning other cases without additional indications on what these cases may be? / P7L17 "The determination (if possible)... (t) follows then" / P7L20 "events" / P7L23 "will increase with more pronounced flood peaks"*

Ok, thanks.

*R3 – Comment 10) P4L11 Please briefly indicate here how the separation is made (I think it is mentioned later)*

Several methods can be applied for hydrograph separation (such as constant-discharge method, concave method, etc). This was the reason why we don't give more indications about this procedure in this section devoted to the modelling approach. The method used in this paper (the constant slope method using the inflection point) is explained in more detailed in the Section 2.3 devoted for the framework application on our study site.

*R3 – Comment 11) P4L20 What is the origin of t? Does it start at t=0, what is the domain of values for t and what is the associated physical meaning? Think of non-specialists readers here.*

This question was also pointed out by Reviewer #2 (Comment 6) and more details description of the solution domain and the impulse response function described by Hayami analytical solution will be added. Please see our response to the Comment 6 of the Reviewer 2.

*R3 – Comment 12) P5L24 This phrase on volume conservation seems a bit strange here. Isn't it guaranteed by the assumption that flow is incompressible?*

Yes it is, (but only in the case of non-reactive solutes, as it is mentioned in the following sentence – P5L25). This sentence (P5L24) will be removed in the new version of the manuscript and P5L25 will be modified as follows: "*We propose here to assess lateral mass fluxes (defined by Eq. 15) during flood, applying the DW model accounting for lateral exchanges, as a transfer function. Thus, the analytical solution of Moussa (1996) is used to resolve the conservative solute mass transfer respecting the mass conservation law and accounting for uniformly distributed lateral fluxes.*"

Study site

*R3 – Comment 13) P8L17 "15-minute" / P9L7 I think you should rather state that "CM and DM play similar roles as CQ and DQ" / P9L20 "the lower the peak flow intensity" / P9L23 "could be reversed"*

Ok.

*R3 – Comment 14) P9L3 What makes you think the selected event is representative? More generally, how do you define a representative event? The question may seem tricky but the answer depends on what you aim to show with the sensitivity analysis you propose. In fact, I do not think you need to assume there are any representative events as soon as the aim of your sensitivity analysis is to prove your methods are "discriminating" enough, meaning your model is able to identify the key factors in the description of the system.*

We agree with this comment and we propose the following modification for the new version of the manuscript:

P9L2 to P9L4 replaced by: "To illustrate the model behaviour described theoretically in Section 2 and to help to define a parametrization strategy, this section presents a sensitivity analysis on a benchmark flood event. This event was defined in order to have similar characteristic (same range of magnitude and parameter's values) than those presented in the model application."

Results

*R3 – Comment 15) P10L6 "but that it was due lateral" / P10L9 "were half these of" / P10L12 "dynamics" / P10L13 "were simulated" / P10L24 "2.6 times higher" / P11L16 "the distribution of values for the model parameters" / P11L17 "intensities, with the aim to retrieve information on flow dynamics" instead of the existing phrasing? P11L18 "contain" rather than "furnish" / P11L20 "which span both the unsaturated and the saturated zones" / P11L24 "did not find" / P11L28 "again with high values only for events" / P12L2 "trends" rather than "patterns" / P12L3 "for the hydrological" / P12L8 "therefore deliver"*

Ok.

*R3 – Comment 16) P10L28 You mention a lower slope but is the difference between 0.3 and 0.4 really striking?*

We will effectively lightly temper this sentence in the new version as follows:

P10L28 will be replaced by: "In contrast, for reach R2 a slightly lower slope was found for mass flux than for water flow (0.3 against 0.4), meaning that lateral inflow was probably a little less mineralized than input flow from station s2."

*R3 – Comment 17) P11L7 Delete "show"?*

Ok. To write it more clearly, we will correct: P11L7 modified as: "*show maxima close to … indicating …*"

*R3 – Comment 18) P11L14 "towards" does not seem appropriate here*

"a*long*" should be more appropriate than "*towards*".

*R3 – Comment 19) P11L21 You indirectly mention the role of antecedent/initial conditions. I think this point should be better and more explicitly addressed as soon as you intend to provide a rather generic of the hydrological behaviour of the system. In particular, do you think you have enough events to draw general conclusions regarding the sensitivity of the model to its initial conditions. This would be a step towards more genericity and a possible way to endow the model with increased predictive capabilities.*

We present in the section "Field monitoring and data processing" how events are selected and how they are sorted in function of the decreasing baseflow at the system outlet. We think that the selected events presented in the paper - typified by various initial baseflow condition and flood flow intensities – allow characterizing both low flow and high flow periods. Thus the selected events are considered relevant to attempt a description of the karst system behaviour as discussed in the Section 5.2. It will be specified in the revised version of the manuscript in P13L29 as follows: "*The selected events are typified by various initial baseflow condition and flood flow intensities allowing thus to characterize both low flow and high flow periods providing a rather generic of the hydrological behaviour of the system.*".

*R3 – Comment 20) P12L11 Does this hypothesize no effect exist on DQ and/or does this assumes DQ is definitely a second-order term in such cases? If so, this could be voiced here.*

In the hypothesis of the calculation of the U value, we only use the $C_Q$ parameter because it characterizes the propagation along the conduit. In order to clarify this section, P12L6 will be modified as follows: "*… in the $C_Q$ parameterization - characterizing the propagation of the flood peak along the 2 reaches R1 and R2.*".

*R3 – Comment 21) P12L16 Where does this approximation come from?*

This approximation comes from the calculated U value based on the total length of the main conduit network. A sentence will be added to make this approximation clearer for readers. The sentence will be modified in the revised version of manuscript as follows: "*Based on the calculated U value and with an estimated total length of 5.4 km for R2, the saturated conduit length is approximatively 1.5 to 2.2 km.*"

Discussion and conclusion

*R3 – Comment 22) This section should be separated into "discussion" and "conclusion". Moreover, I tend to disagree with the rather atypical names of the sections in this paper. According to me, "Study site" should be the 2.2 part of a Section 2 named "Material and methods", following the 2.1 part entitled "Modelling approach". I found the discussion repetitive in some parts, which I suggest to remove. This would facilitate the reader's task and make the additional indications (i.e. the real discussion elements) more punchy.*

Thanks for your advices. The "discussion & conclusion" will be divided in the aim to facilitate the reader throughout the paper. We also agree that the discussion can be repetitive in some parts and following your recommendation some sections will be reduced in the new version of the manuscript such as *P12L20 to P13L2 and P13L31 to P13L34*. Regarding the Modelling approach and Study site sections, we prefer to keep these two sections as independent sections, because the main aim of this paper is to present the model, which appears better highlighted in a separated section.

*R3 – Comment 23) P13L18-19 What would be the effect of a smaller spatial discretization in the streamwise direction? Could the model handle a smaller dx and could this be part of the discussion?*

This is an interesting point. In fact, this would be interesting to discretize the main conduit in more reaches to have a better description of the spatial variability of the lateral exchanges along the network. The model is generic and a smaller dx can easily be handled. However if dx is small, the time step dt must be taken also small in order to have exact integral calculations in Eqs 8, 9 and 19. Depending on the size of the spatial discretization, tributaries could also be taken into account along the specific reach. However, the application of the inverse model on numerous smaller reaches requires a large set of monitoring stations that are not always accessible in karst systems. In this way, Moussa (1997) proposed a flood routing model based on the Hayami analytical solution which also takes into account the geomorphological structure of the catchment. The coupling of two knowledges (hydrographs and topographic elevations) gives then interesting information to describe the lateral exchanges along a river stream. In karst conduit system, a smaller spatial discretization of the lateral exchange in a karst conduit should be also coupled with others knowledges where the topographic elevations may not always be relevant to describe flood routing propagation, specifically in the saturated zone.

Additional comments will be included in the discussion of the "Modelling Framework" in P13L28 as follows: *"... with highly variable lateral contributions. An interesting point would be to discretize the conduit network in smaller reaches. Nevertheless, it should require a large set of monitoring stations that are not always accessible in karst systems. In this way, Moussa (1997) proposed a methodology to identify the transfer function based on the Hayami analytical solution using the topological elevations of a distributed channel network flood routing modelling. However, in karst conduit system, besides the partial access to the conduit, the karst conduit elevations may not always be relevant to describe flood routing propagation, specifically in the saturated zone.".*

*R3 – Comment 24) P13L29 Alternatively what are the requirements for boundary conditions not to be poor? What are the required BC to ensure the model performs well (or to make sure the BC are not the weak point of the description)?*

We think that it is a large issue, not restricted to karst media, as showed by Szeftel et al. (2011) for stream channel in a watershed. The only way to ensure the model performs well is to monitor the lateral exchanges in order to compare it to simulations. A first step was proposed by Charlier et al. (2015) to compare the hydrograph dynamics of lateral springs with the simulated lateral exchanges.

*R3 – Comment 25) Delete most of P12L20 to P13L2 / P13L4 "variability" is repeated / The discussion P13L7-16 is very instructive / Delete P13L31-34 and start the next phrase with "The functional scheme..." / P14L3 "Fig" not "fig" and in other occurrences too / P14L6 "had lateral inflows" / P14L7 "these mineralized inputs" / P14L11 "mineralized" / P14L19 Delete "derived" / P14L31 "R1" / P15L1-7 Make it shorter or delete / P15L8+ The last part of the document is convincing and well-written. Should be the conclusion and the message for future research leads.*

We thank you for your advice and correction about English formulation. These points will be modified in the revised version of the manuscript.

**Author comment to Referee #4**

*The manuscript introduces a new framework which enables assessment of inflows/outflows to/from the channel reach in a karst aquifer. The paper presents a nice combination of field and modelling work and is as such worth publishing. My comments are mostly related to the presentation of concepts (and to some extent results) which should be made clearer.*

Thank you!

*Here is the list of comments (P#L#) refers to the page and line number to which the comment refers to the page and line number the comment refers to:*

*R4 – Comment 1) The concept of Diffusive wave equation was first introduced at P2L35 and reintroduced at P3L5. Restructure these paragraphs.*

Ok, a restructuration of these two paragraphs in the Introduction Section will be performed in the revised version of the manuscript.

*R4 – Comment 2) P3L8 A statement that DW is used for mass transport is a bit misleading. Although the equation is the same as the ADE, it is built on different conservation principles and driving forces. The statement might confuse a reader. The same comment goes to P327.*

We agree that the formulation is awkward. In fact, the diffusive wave equation and the advection-diffusion transport equation have very similar mathematical equation, but of course do not describe the same processes. Additional details will be added to show the differences in the physical basis of each of the diffusive wave equation and the advection-diffusion transport equation. The same comment has been done by the Referee #2 (Comment 5). These sentences will be clarified in the revised version of the manuscript.

*R4 – Comment 3) P4L15: What do you mean with without downstream boundary conditions. Please clear up.*

We agree with this comment also done by Referee #2 (Comment 6), and will add adequate explanation in the text concerning the initial condition ($t = 0$; $Q(x,t) = 0$) and the two boundary conditions: i) upstream for $x = 0$ $Q(x,t) = \delta(t)$ (Dirac function) and for $x = X \rightarrow \infty$ $Q(x, t)=0$. Please see our response to the Comment 6 of the Reviewer 2.

*R4 – Comment 4) P4L 14-17: This paragraph is somehow awkward. What do you mean with direct model? The $Q_{I,routed}$ is introduced, but not told what it represents; this would be helpful for someone not familiar with the older literature... The aim of 2.1.1. is somehow lost until P4L21: I miss an earlier clear statement that $C_Q$ and $D_Q$ are looked for. How are $Q_{I,routed}$ and $Q_{O,flood}$ compared?*

This paragraph will be modified in the revised version of the manuscript in order to clarify all terms described in the modelling section. The term "direct model" refers to the model where the inputs are the upstream hydrograph, lateral inflow/outflow and the model parameters (C and D), and the output a simulated hydrograph downstream. In the opposite the "inverse

model" refers to the model where the inputs are upstream and downstream hydrographs and the model parameters, and the output the lateral inflow/outflow.

Besides, more detailed information is given in the description of the framework (Section 2.3.). P4L21 will be moved after Equation (4) and modified as follow: "*The direct model is then used to compare $Q_{l,routed}(t)$ to $Q_{O,flood}(t)$ and perform the parameterization of $C_Q$ and $D_Q$ as described afterward (Section 2.3 & 3.3.2).*".

*R4 – Comment 5) P5L8 (and before): It looks like that the model allows assessment of q(x,t), however as far as I understanding it allows only time dependence of the total in/out-flow between I and O. In other words, spatial distribution of in/out-flows is not obtained?*

We totally agree with this comment. The equation (9) is written incorrectly. The upper limit of Eq.(9) is not time, but the length of the reach L. This error will be corrected in the new version of the manuscript as follow:

$$Q_{A,flood}(t) = \int_0^L q(x,t)dx$$

*R4 – Comment 6) P5EQ10 and EQ11: Change $A_Q(t)$ to $Q_A(t)$.*

Eq 10 and Eq11 are correctly written. $Q_A(t)$ is calculated in a second step, according to Eq. 12, 13 and 14.

*R4 – Comment 7) P5L20-25: The paragraph is hard to read. 1) What do you mend by application of ADE is not so obvious. Does this mean that equation itself is not obvious or that the solution is not straightforward. ... If I understand correctly, the idea is to determine lateral flow from DW (previous chapter) and introduce it into ADE to finally get M(t) and S(t). Why introducing Q(t) in Equation 15 ? Is this $Q_A(t)$ ? I miss clear presentation of ideas and concepts. It would be good if you put in the ADE equation.*

We agree that some terms used are confusing, and we will give complementary explanation as already shown in our response to Reviewer #2 Comments 4 and 5. In fact, the application of ADE equation is not straightforward because it is usually applied on concentration evolution during steady flow condition in which mass conservation law is respected. However, in the case of unsteady flow condition, the ADE was applied on solute concentration. The equation 15 includes Q(t) in the aim to attempt the mass conservation law to be respected in the case of unsteady flow conditions. The sentence P5L21 will be reformulated as "*ADE is not straightforward*".

*R4 – Comment 8) P9L12-20: Can one infer from DW equation that sensitivity to celerity is orders of magnitude higher than sensitivity to diffusivity?*

Yes, it is for the application done on a benchmark flood event to our case study. The sensitivity analysis of the diffusive wave equation shows that celerity (which multiplies the first order derivative of Q) is in fact more sensitive than diffusivity (which multiplies the second order derivative of Q).

**REFERENCES**

Charlier, J.-B., Moussa, R., Bailly-Comte, V., Danneville, L., Desprats, J.-F., Ladouche, B., and Marchandise, A.: Use of a flood-routing model to assess lateral flows in a karstic stream: implications to the hydrogeological functioning of the Grands Causses area (Tarn River, Southern France), environmental Earth Sciences, 74, 7605–7616, doi:10.1007/s12665-015-4704-0, 2015.

Chow, V.T.: Open-Channel Hydraulics. McGraw Hill, New York, 680 pp, 1959.

Le Moine, N., Andréassian, V., and Mathevet, T.: Confronting surface- and groundwater balances on the La Rochefoucauld-Touvre karstic system (Charente, France), Water Resour. Res., 44, W03403, doi:10.1029/2007WR005984, 2008.

Moussa, R. and Bocquillon, C.: Algorithms for solving the diffusive wave flood routing equation, Hydrol. Process., 10, 105–123, doi:10.1002/(SICI)1099-1085(199601)10:1<105::AID-HYP304>3.0.CO;2-P, 1996.

Moussa, R. (1997), Geomorphological transfer function calculated from digital elevation models for distributed hydrological modelling. Hydrol. Process., 11: 429–449. doi:10.1002/(SICI)1099-1085(199704)11:5<429::AID-HYP471>3.0.CO;2-J

Szeftel, P., (Dan) Moore, R., and Weiler, M.: Influence of distributed flow losses and gains on the estimation of transient storage parameters from stream tracer experiments, Journal of hydrology, 396, 277–291, doi:10.1016/j.jhydrol.2010.11.018, 2011.

---

## Author Response (AR2)

**Responses to reviewer's comments**

(Hydrol. Earth Syst. Sci. Discuss., doi:10.5194/hess-2016-565, 2016)

We are very grateful to the Editor Mauro Giudici and the reviewers for their constructive comments on the manuscript. Please find below our responses and the detailed modification (in blue) to the comments (in black) of Editor and the two reviewers. All modifications in the text are highlighted in yellow in the new version of the manuscript.

**Editor report**

The paper requires a minor revision, mostly related to some linguistic aspects and with the proper use of scientific terms.
I expect that the paper can be revised to improve the English language and to have a more appropriate use of terms like "forward and inverse problem", "inverse modelling", etc.

All recommendations made by the Referee 2 about the use of appropriate terms concerning models and inverse approaches were considered in the new version of the manuscript.

**Author comment to Referee #2**

I would like to find out that practically all my critical remarks presented previously were taken into account and respective comments and corrections were introduced by the Authors into the revised version of submitted manuscript.
However, although in the same review I tried to pay the Authors' attention that the subject of their manuscript in fact deals with solution of the inverse problem, in many places of the considered text, including even its title, they use still the term "inverse model". To my mind the term "model" simply means the equation or the system of equations describing the analyzed phenomenon. In the considered case both models are constituted by the partial differential equations of parabolic type: the advection – diffusion equations. If in assumed region of solution and for the auxiliary conditions imposed at its limits these equations are solved then it is said that their direct solutions was performed. In the case when knowing the solution we are looking for some auxiliary conditions or for the parameters describing the considered process then it is said that the inverse problem is solved. In my opinion this is the case considered by the Authors because they consider lateral inflow/outflow as unknown. Therefore, instead of the term "inverse model" they should rather use the term "inverse problem for the advection - diffusion equations" as in the mathematical physics the terms "direct models" and "inverse models" do not exist. The Authors should not introduce new nomenclature if it is not necessary (page 3, lines 15-20).
We thank you for your advice about the more appropriate terminology. In order to clarify the formulations all along the manuscript, the nomenclature was modified by deleting the terminology "direct model" and by substituting the term "inverse model" by "inverse problem (for advection-diffusion equations)".

My second remark is dealing with Eq. (4). In explanation given below this equation it is stated that dT is "the time step of integration". In definition of integral "the step of integration" does not exist and dT has another interpretation (please see definition of integral in any book). The time step of integration will occur when the convolution integral (4) is replaced using the respective approximating formula.

Ok. We understand the remark about the formulation used for *dT* and we prefer thus to remove it in the submitted version of the manuscript.

**Author comment to Referee #3**

I am totally satisfied with this version of the paper.

Thanks.

---

## Author Response (AR3)

**Response to the Editor**

(Hydrol. Earth Syst. Sci. Discuss., doi:10.5194/hess-2016-565, 2016)

**Editor report**

*The manuscript has been improved, but some further technical modifications can be useful to facilitate a proper editorial finalization of the paper.*

We thank you for your recommendations. They were considered in the new version of the manuscript. The modifications are highlighted in yellow in the text.

---

## Author Response (AR4)

**Response to the Editor**

(Hydrol. Earth Syst. Sci. Discuss., doi:10.5194/hess-2016-565, 2016)

**Editor report**

*The paper can be published provided a few technical corrections are introduced. Since they are not interesting from the scientific point of view, I list them in the non-public comments.*

We thank you for your comments. They were considered in the new version of the manuscript. Nevertheless, we prefer to keep the structure of equations (5) and (20), because the exponential appears clearer for readers. The modifications are highlighted in yellow in the text.